# Acetylation-mediated remodeling of the nucleolus regulates cellular acetyl-CoA responses

Ryan Houston[1], Shiori Sekine[1,2], Michael J. Calderon[3], Fayaz Seifuddin[4], Guanghui Wang[4], Hiroyuki Kawagishi[4], Daniela A. Malide[4], Yuesheng Li[4], Marjan Gucek[4], Mehdi Pirooznia[4], Alissa J. Nelson[5], Matthew P. Stokes[5], Jacob Stewart-Ornstein[6], Steven J. Mullett[7], Stacy G. Wendell[7], Simon C. Watkins[3], Toren Finkel[1,2,4], Yusuke Sekine[1,4,8]*

1 Aging Institute, Department of Medicine, University of Pittsburgh, Pittsburgh, Pennsylvania, United States of America, 2 Division of Cardiology, Department of Medicine, University of Pittsburgh, Pittsburgh, Pennsylvania, United States of America, 3 Department of Cell Biology, Center for Biologic Imaging, University of Pittsburgh, Pittsburgh, Pennsylvania, United States of America, 4 National Heart, Lung, and Blood Institute, NIH, Bethesda, Maryland, United States of America, 5 Cell Signaling Technology, INC., Danvers, Massachusetts, United States of America, 6 Department of Computational and Systems Biology, University of Pittsburgh and Hillman Cancer Center, Pittsburgh, Pennsylvania, United States of America, 7 Department of Pharmacology and Chemical Biology, the Health Sciences Metabolomics and Lipidomics Core, University of Pittsburgh, Pittsburgh, Pennsylvania, United States of America, 8 Division of Endocrinology and Metabolism, Department of Medicine, University of Pittsburgh, Pittsburgh, Pennsylvania, United States of America

* SEKINEY@pitt.edu

**Data Availability Statement:** All relevant data are within the paper and its Supporting Information files.

## Abstract

The metabolite acetyl-coenzyme A (acetyl-CoA) serves as an essential element for a wide range of cellular functions including adenosine triphosphate (ATP) production, lipid synthesis, and protein acetylation. Intracellular acetyl-CoA concentrations are associated with nutrient availability, but the mechanisms by which a cell responds to fluctuations in acetyl-CoA levels remain elusive. Here, we generate a cell system to selectively manipulate the nucleo-cytoplasmic levels of acetyl-CoA using clustered regularly interspaced short palindromic repeat (CRISPR)-mediated gene editing and acetate supplementation of the culture media. Using this system and quantitative omics analyses, we demonstrate that acetyl-CoA depletion alters the integrity of the nucleolus, impairing ribosomal RNA synthesis and evoking the ribosomal protein-dependent activation of p53. This nucleolar remodeling appears to be mediated through the class IIa histone deacetylases (HDACs). Our findings highlight acetylation-mediated control of the nucleolus as an important hub linking acetyl-CoA fluctuations to cellular stress responses.

## Introduction

Intracellular metabolites dynamically fluctuate in living organisms according to the availability of their source nutrients such as glucose, lipids, and amino acids. Cells employ a variety of molecular machineries to monitor the concentration of these metabolites. During metabolic

**Funding:** This work was supported by National Institutes of Health (https://www.nih.gov/) (Intramural Funds and 1 R01 HL142663 to T.F.), Fondation Leducq (https://www.fondationleducq.org/)(to T.F.), the Aging Institute of the University of Pittsburgh (https://ai.dom.pitt.edu/)(to Y.S. and S.S.). the UPMC Competitive Medical Research Fund (https://www.oorhs.pitt.edu/research-funding/competitive-medical-research-fund-grant-application) (to Y.S. and S.S.), National Cancer Institute (https://www.cancer.gov/) (4R00CA297727 to JS-O), National Institutes of Health (https://www.nih.gov/)(S10OD023402 to SGW). The funders had no role in study design, data collection and analysis, decision to publish, or preparation of the manuscript.

**Competing interests:** The authors have declared that no competing interests exist.

**Abbreviations:** acetyl-CoA, acetyl-coenzyme A; ACLY, ATP citrate lyase; ACSS2, acyl-CoA synthetase short chain family member 2; ActD, actinomycin D; AOBS, Acousto-Optic Beam Splitter; ASA-KO, acetate-supplemented ACLY knockout; ASA-WT, acetate-supplemented ACLY wild type; ATP, adenosine triphosphate; CoA, coenzyme A; CDS, coding sequence; CRISPR, clustered regularly interspaced short palindromic repeat; dFBS, dialyzed FBS; DGLA, Dihomo-γ-linolenic acid; FBL, fibrillarin; FBS, fetal bovine serum; FC, fold change; FDR, false discovery rate; FRAP, fluorescence recovery after photobleaching; FUrd, 5-Fluorouridine; GO, gene ontology; HDAC, histone deacetylase; HPLC, high performance liquid chromatography; IDR, intrinsically disordered region; IL, interleukin; KO, knockout; LC-MS, liquid chromatography-mass spectrometry; LC-UV, liquid chromatography with UV detection; LLPS, liquid–liquid phase separation; logCPM, log$_2$ counts per million; MEFs, mouse embryonic fibroblasts; MEM, Minimum Essential Medium; mGFP, monomeric GFP; mRNA, messenger RNA; mTORC1, mechanistic target of rapamycin complex 1; NAD, nicotinamide adenine dinucleotide; NCL, nucleolin; NCL-mGFP, monomeric GFP-tagged NCL; NPM1, nucleophosmin 1; PA, 3-picolylamide; PSMs, peptide-spectrum matches; PTMs, post-translational modifications; rRNA, ribosomal RNA; sgRNA, single guide RNA; SGs, stress granules; siRNA, small interfering RNA; TCA, tricarboxylic acid; TMM, trimmed mean of M-values; TMT, tandem mass tag; tRNA, transfer RNA; UBF, upstream binding factor; UDP-N-acetylglucosamine, uridine diphosphate N-acetylglucosamine; WT, wild-type; YFP, yellow fluorescent protein.

fluctuations, these machineries activate signaling pathways that modulate gene expression and protein function to maintain metabolic homeostasis. The molecular links between the metabolite sensing and the subsequent cellular responses have been extensively studied and several dedicated molecular networks have been described [1–3]. However, partly due to experimental difficulties to selectively manipulate target metabolites among highly interconnected metabolic networks, the mechanisms for cellular primary responses toward fluctuations for most metabolites are not understood.

Acetyl-coenzyme A (acetyl-CoA) is a central metabolite that integrates diverse nutritional inputs into the biosynthesis of essential biomaterials including adenosine triphosphate (ATP), fatty acids, and steroids, and therefore can be viewed as a critical indicator of the cellular metabolic state [4,5]. In addition, acetyl-CoA provides an acetyl donor for lysine acetyltransferases catalyzing protein acetylation, which modulates biophysical properties of modified proteins, thereby altering their protein stability, enzymatic activity, or interactions with other proteins. This modification is reversed by lysine deacetylases. Hence, in accordance with the nutrient availability and the cellular metabolic state, protein acetylation can reversibly regulate a variety of biological processes including gene expression, signal transduction pathways, and metabolic flux [6,7].

In mammalian cells, acetyl-CoA is compartmentalized into a mitochondrial pool and a nuclear/cytoplasmic pool [4,8]. In the mitochondrial matrix, acetyl-CoA is generated by the metabolism of nutrients including glucose, fatty acids, and amino acids. Mitochondrial acetyl-CoA can enter the tricarboxylic acid (TCA) cycle, thereby generating ATP and reducing equivalents [e.g., a reduced nicotinamide adenine dinucleotide (NADH)], or it can be utilized for the acetylation of mitochondrial proteins. Since acetyl-CoA is membrane impermeant, there is no direct exchange between mitochondrial acetyl-CoA and the acetyl-CoA in the cytosol. Rather, the TCA intermediate citrate can be exported from the mitochondria to the cytosol where it can become the predominant source for cytoplasmic and nuclear acetyl-CoA. The mitochondria-exported citrate is converted to acetyl-CoA and oxaloacetate by the enzyme ATP citrate lyase (ACLY) [9]. This nucleo-cytoplasmic acetyl-CoA pool serves as a building block for lipid synthesis including fatty acids, cholesterol, and steroids, and for synthesis of nucleotide sugars including uridine diphosphate (UDP)-N-acetylglucosamine. It also serves as an acetyl donor for acetylation of cytosolic and nuclear proteins including histones. Another source of nucleo-cytoplasmic acetyl-CoA comes from the metabolite acetate, which is derived from various extra and intracellular processes such as food digestion, gut microbial metabolism, alcohol oxidation, and the deacetylation of acetylated proteins [10–14]. In addition, glucose-derived pyruvate can be directly converted to acetate under certain conditions [15,16]. Cytosolic and nuclear acetate is utilized for acetyl-CoA synthesis through a reaction catalyzed by the enzyme acyl-CoA synthetase short chain family member 2 (ACSS2). In contrast to whole body *ACLY*-deficient mice that are embryonic lethal, *ACSS2*-deficient mice have no apparent phenotype under normal breeding conditions [17,18]. This would support the predominant role of the citrate-ACLY pathway for the nucleo-cytoplasmic acetyl-CoA production during development, and potentially under other circumstances as well. However, accumulating evidence has indicated that in tumor cells, in cells under metabolic stress such as hypoxia, and in certain types of cells such as neurons, T cells, and hepatocytes, the acetate-ACSS2 pathway can play a critical role in cell proliferation, lipid synthesis, and acetylation of histones and non-histone proteins [11,12,18–26]. Moreover, it has been demonstrated that *ACLY*-deficient mouse embryonic fibroblasts (MEFs) and LN229 human glioblastoma cells exhibit up-regulation of ACSS2 and that exogenously added acetate can be utilized for acetyl-CoA production in these cells [27]. These observations indicate a coordinated relationship between these 2 pathways in order to ensure the requisite supply of acetyl-CoA in the nucleo-cytoplasmic compartment.

Intimate connections between nucleo-cytoplasmic acetyl-CoA levels, the status of protein acetylation and cellular responses have been illustrated by multiple recent studies. In yeast cells grown under glucose-limited conditions, oscillations of acetyl-CoA are observed in accordance with distinct metabolic phases, and an increase in acetyl-CoA is highly correlated with acetylation of transcriptional coactivators and histones regulating growth genes [28]. Also, in mammalian cell models and mouse tissues, acetyl-CoA levels and the ratio of acetyl-CoA to coenzyme A (CoA) can be critical determinants of the status of histone acetylation [29,30]. Thus, alterations in acetyl-CoA abundance by modulating its source nutrients and acetyl-CoA producing enzymes, i.e., ACLY or ACSS2 affect transcription through histone acetylation. Recent findings have indicated that acetyl-CoA locally produced in the nucleus by nuclear targeted ACLY or ACSS2 contributes to site-specific histone acetylation, which specifies gene induction and chromatin remodeling in a context dependent manner [22,23,26,29,31–33]. Moreover, nutrient deprivation or starvation causes a rapid decline in acetyl-CoA levels in cultured cells and in some mouse tissues, which is accompanied by deacetylation of proteins [29,34]. In mammalian cells and yeast, pharmacological and genetic manipulations that deplete cytosolic acetyl-CoA induce autophagy [34,35]. Acetylation of multiple autophagy-related proteins plays a crucial role for the autophagy induction [34,36–39]. As such, acetyl-CoA fluctuations appear to influence various biological responses through alterations of protein acetylation.

Here, we have developed a cell line that lacks ACLY and whose nucleo-cytoplasmic acetyl-CoA production is thereby solely dependent on exogenously supplemented acetate. Removal of acetate from the culture media enables us to rapidly deplete intracellular acetyl-CoA. Using this cell line, we provide quantitative data sets for alterations in messenger RNA (mRNA) levels, protein abundance and acetylation status in cells experiencing acute acetyl-CoA depletion. These analyses uncovered that in response to acetyl-CoA depletion, the integrity of the nucleolus, an organelle critical for ribosome biosynthesis, was morphologically and functionally remodeled. This nucleolus remodeling leads to the up-regulation of the stress responsive transcription factor p53 in a ribosomal protein–dependent manner. Furthermore, these alterations were found to be suppressed by class IIa histone deacetylase (HDAC) inhibitors. We identified multiple nucleolar proteins whose acetylation levels were potentially regulated by the class IIa HDACs, suggesting that class IIa HDAC-dependent deacetylation of nucleolar proteins may play an important role in regulating the integrity of the nucleolus and the induction of a nucleolus-dependent stress response when acetyl-CoA levels decline.

## Results

### Acetate-dependent control of acetyl-CoA production in ACLY KO cells

In order to understand cellular responses to fluctuations in nucleo-cytoplasmic acetyl-CoA abundances, we sought to devise a cell system where we could selectively and robustly control acetyl-CoA levels. To achieve this, we focused on establishing a cell line whose acetyl-CoA in the nucleo-cytoplasmic compartment was solely synthesized through the acetate-ACSS2 axis, assuming that such a cell line would enable us to manipulate acetyl-CoA levels by simply modulating the amount of acetate exogenously supplemented in the culture media (**Fig 1A**). Hence, using the clustered regularly interspaced short palindromic repeat (CRISPR)-Cas9 system, we targeted the *ACLY* gene in HT1080 human fibrosarcoma cells to make *ACLY*-deficient cell lines. A plasmid coding an *ACLY*-targeting single guide RNA (sgRNA) together with Cas9 was transfected into HT1080 cells, and the transfected cell clones were recovered either in the standard culture media (0-mM added acetate) or in the same media but with excess amounts (2 or 20 mM) of sodium acetate (**S1A Fig**). Among the cell clones screened, putative *ACLY*

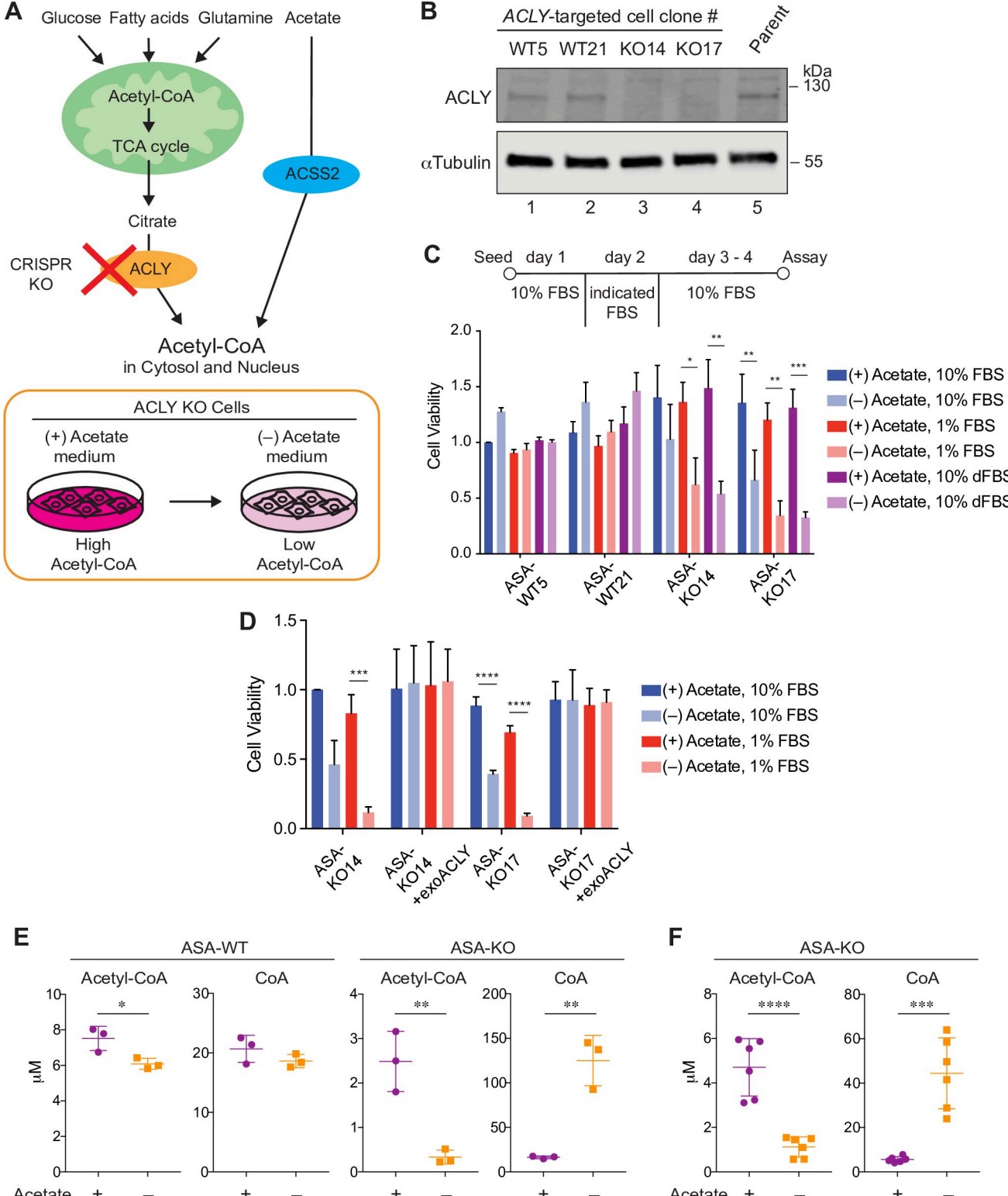

**Fig 1. Acetate dependency for acetyl-CoA production in ASA-KO cells.** (A) Schematic representation of the pathways for nucleo-cytoplasmic acetyl-CoA production in mammalian cells (top) and of the acetate-dependent control of acetyl-CoA in ACLY KO cells (bottom). (B) Immunoblotting for ACLY in representative ACLY-targeted cell clones maintained in 20-mM acetate-supplemented media and in parental HT1080 cells. αTubulin is shown as a loading control. (C) Cell viability assay for ASA-WT and KO cells cultured in the presence or absence of 20-mM acetate in the culture media with indicated FBS conditions over 4 days. On day 2, the original medium containing 10% FBS was replaced for 24 hours with media containing either 10%

FBS, 1% FBS, or 10% dFBS. Data are shown as mean ± SD ($n$ = 3 independent experiments). $^*P < 0.05$, $^{**}P < 0.01$, $^{***}P < 0.001$ (1-way ANOVA followed by Tukey multiple comparisons test). (D) Cell viability assay for ASA-KO cells with or without exoACLY using the same experimental procedures as in "C." Data are shown as mean ± SD ($n$ = 3 independent experiments). $^{***}P < 0.001$, $^{****}P < 0.0001$ (1-way ANOVA followed by Tukey multiple comparisons test). (E) LC-UV quantification of acetyl-CoA and CoA in cell lysate from ASA-WT and ASA-KO cells cultured for 4 hours in media containing 1% FBS with or without 20-mM acetate. Estimated single cell concentrations are shown as mean ± SD ($n$ = 3 biological replicates). $^*P < 0.05$, $^{**}P < 0.001$ (Unpaired $t$ test). (F) LC-MS quantification of acetyl-CoA and CoA in cell lysate from ASA-KO cells cultured as in "E." Estimated single cell concentrations are shown as mean ± SD ($n$ = 6 biological replicates). $^{***}P = 0.0001$, $^{****}P < 0.0001$ (Unpaired $t$ test). The data underlying the graphs in Fig 1 can be found in S1 Data. acetyl-CoA, acetyl-coenzyme A; ACLY, ATP citrate lyase; ACSS2, acyl-CoA synthetase short chain family member 2; ASA-KO, acetate-supplemented ACLY knockout; ASA-WT, acetate-supplemented ACLY wild type; CoA, coenzyme A; CRISPR, clustered regularly interspaced short palindromic repeat; dFBS, dialyzed FBS; exoACLY, exogenous expression of ACLY; FBS, fetal bovine serum; KO, knockout; LC-MS, liquid chromatography-mass spectrometry; LC-UV, liquid chromatography with UV detection; TCA, tricarboxylic acid.

genome-edited clones were found to only exist in the acetate supplemented media (S1B Fig), suggesting that acetate supplementation increased the viability or growth of *ACLY*-deficient cells. This observation was consistent with the previously reported phenotype of *ACLY*-deficient MEFs [27]. We selected 2 independent *ACLY*-deficient [knockout (KO)] and *ACLY* expressing [wild-type (WT)] clones, respectively, that henceforth were continuously cultured in the presence of 20-mM acetate for further analyses (Fig 1B and S1C Fig). Hereafter, we refer to these cells as a̲cetate-s̲upplemented *A̲CLY* K̲O̲ (ASA-KO) and a̲cetate-s̲upplemented *A̲CLY* W̲T̲ (ASA-WT), respectively. In this culture condition, ASA-KO cells did not exhibit any defect in viability when compared with ASA-WT cells [Fig 1C, (+) Acetate, 10% fetal bovine serum (FBS)]. However, removal of acetate from the culture media for 4 days significantly impaired the viability of ASA-KO but not ASA-WT cells (Fig 1C). Consistent with previous observations that FBS contains submillimolar concentration of acetate [27,40], either a decrease in FBS concentration from 10% to 1% or a use of 10% dialyzed FBS (dFBS) enhanced the vulnerability of ASA-KO cells, while these manipulations had no significant effect on viability in the acetate-supplemented conditions (Fig 1C). Moreover, exogenous expression of ACLY in ASA-KO cells rescued the sensitivity to acetate removal, indicating that the acetate dependence of these cells was caused by ACLY deficiency (Fig 1D).

Using this cell system, we next examined whether acetate removal modulated intracellular acetyl-CoA levels in ASA-KO cells. We used culture media containing 1% FBS during acetate removal to maximize the effect of acetate withdrawal, although similar, albeit slightly milder effects were seen following acetate withdrawal in the setting of 10% FBS. Quantifications of acetyl-CoA and CoA in whole cell lysates using high performance liquid chromatography (HPLC) with UV detection (LC-UV) demonstrated that withdrawing acetate for 4 hours resulted in a more than 80% decrease in acetyl-CoA levels in ASA-KO cells (Fig 1E and S2 Fig). This intervention had a much milder effect (approximately 20% decrease) on acetyl-CoA levels in ASA-WT cells (Fig 1E and S2 Fig). These results suggest that exogenously added acetate is a predominant source for intracellular acetyl-CoA in ASA-KO cells, while supplemented acetate only modestly contributes to intracellular acetyl-CoA levels in ASA-WT cells. In contrast, the amount of CoA was drastically increased in ASA-KO cells but not in ASA-WT cells in response to acetate removal (Fig 1E and S2 Fig). These reciprocal alterations in the levels of acetyl-CoA and CoA were also seen in the previously reported models of acetyl-CoA depletion in mammalian cells [29,30], but the degree was more pronounced in our ASA-KO cells. To further confirm these alterations in ASA-KO cells, we also measured the levels of acetyl-CoA and CoA using high-resolution liquid chromatography-mass spectrometry (LC-MS). We detected similar fluctuations in acetyl-CoA (approximately 80% decrease) and CoA (7- to 8-fold increase) using these complementary methods (Fig 1F). Concentrations of acetyl-CoA and CoA in a single ASA-KO cell with acetate, estimated from the LC-MS data, were 4.70 μM and 5.62 μM, respectively, which were within the similar range of concentrations previously

determined in LN229 cells and interleukin 3 (IL-3)-dependent hematopoietic cells using LC-MS [29], as well as concentrations from our LC-UV analysis (**Fig 1E**). Moreover, an approximate 20% and 80% contribution of acetate to intracellular acetyl-CoA in ASA-WT and ASA-KO cells, respectively, is consistent with previous analysis employing stable isotope-labeled acetate in *ACLY* WT and KO MEFs [27]. Taken together, in ASA-KO cells, even in the presence of other nutrient sources such as glucose and lipids, the nucleo-cytoplasmic acetyl-CoA levels are seemingly tunable solely by adding or removing acetate in the culture media.

## Acetyl-CoA depletion modulates protein acetylation

Using the ASA-KO cell system, we profiled global cellular responses following acute acetyl-CoA depletion. As cytoplasmic acetyl-CoA is mainly utilized for lipid synthesis and protein acetylation, we examined whether a rapid decline in acetyl-CoA levels affected these down-stream events. Quantifications of total cholesterol and a series of fatty acids revealed that although some fatty acids including linoleic acid and Dihomo-γ-linolenic acid (DGLA) exhibited significant decreases, overall alterations in total amounts of cholesterol and fatty acids after 4 hours of acetate removal were modest when compared to the dramatic decline in acetyl-CoA levels at the same time point (**Fig 2A** and **Fig 1E** and **1F**). Thus, at least at early time points, dramatic reductions in acetyl-CoA are not fully reflected by marked alterations in total cholesterol and lipid amounts.

In order to examine global changes in the status of protein acetylation upon acetyl-CoA depletion, we conducted acetylome analyses using an acetyl-lysine motif antibody-based immunoaffinity purification. Recovered acetylated lysine-containing peptides were then detected through LC-MS. This approach identified 1,307 acetylated lysine-containing peptides in ASA-KO cells cultured in the presence of acetate (**S1 Table**). Among these, 658 peptides exhibited more than a 50% decrease after 90 minutes of acetate withdrawal {$\log_2$ [fold change (FC): (−) Acetate / (+) Acetate] < −1.0} suggesting that acetyl-CoA depletion caused rapid deacetylation of nearly half of all acetylated proteins (**Fig 2B** and **S1 Table**, column B). These peptides included a wide variety of proteins including molecules involved in transcription, translation, ribosomal RNA (rRNA) processing, mRNA splicing, nucleosome assembly, and cell–cell adherence junctions (**Fig 2C**). By immunoblotting using pan- and site-specific- anti-acetyl-lysine antibodies, we detected immediate and robust decreases in acetylation of both histone and non-histone proteins upon acetate withdrawal in ASA-KO cells (**Fig 2D** and **2E**). Collectively, these findings suggest that unlike the cholesterol or lipid pool, alterations in intracellular acetyl-CoA levels are rapidly reflected in the status of protein acetylation. We note that although it was to a lesser extent when compared with ASA-KO cells, substantial decreases in acetylation of histones and αTubulin were also observed in ASA-WT cells (**S3A Fig**). This may be due to the observed mild decrease in acetyl-CoA levels in ASA-WT cells upon acetate removal (**Fig 1E**). It has also been reported that deacetylation of histones can serve as a supply of acetate under certain conditions [13,14]. As such, deacetylation might ensue as a means to simply restore acetate levels, in a homeostatic cellular response that is largely independent of acetyl-CoA fluctuations.

## Acetyl-CoA depletion modulates gene expressions

Since we observed marked alterations in protein acetylation including on histones, we performed transcriptome analyses by RNA sequencing of ASA-KO cells cultured in the presence or absence of acetate to examine the transcriptional alterations following acetyl-CoA depletion. Consistent with a notion that histone deacetylation suppresses transcription, we identified a large number of transcripts down-regulated 4 hours after acetate withdrawal {526 and 1,096

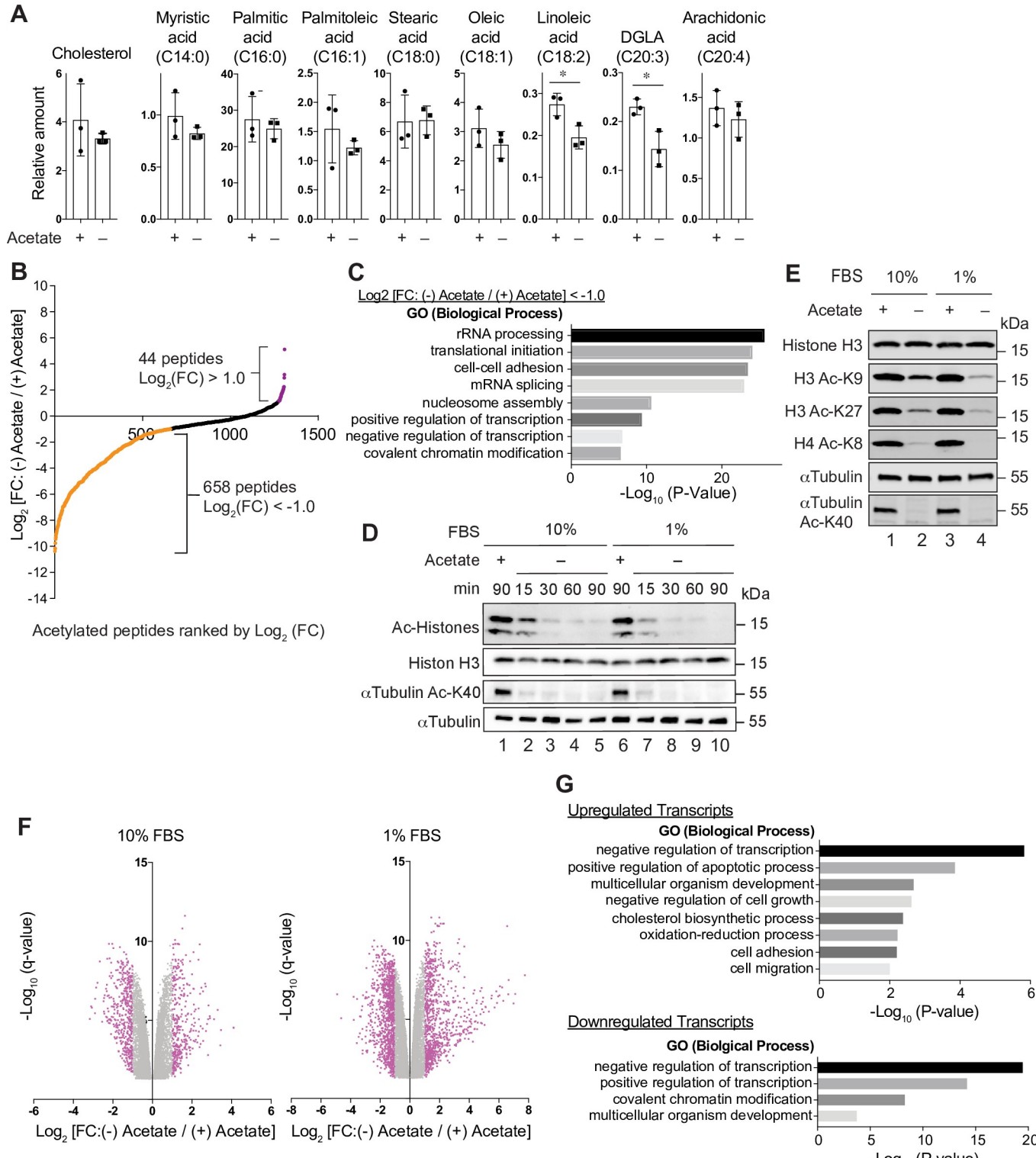

**Fig 2. Alterations in lipid synthesis, acetylation, and gene expression following acetyl-CoA depletion.** (A) Quantification of total cholesterol and fatty acids in ASA-KO cells cultured for 4 hours in 1% FBS containing media with or without 20-mM acetate. Data are shown as mean of 3 biological replicates. *$P < 0.05$ (Unpaired $t$ test). (B) Plot of acetylated peptides ranked by $\log_2$ FC 90 minutes after acetate removal in ASA-KO cells. Values are from S1 Table, column B (mean of 2 independent technical replicates). (C) GO analysis for putative deacetylated peptides identified in the acetylome analysis. Enriched representative biological processes are shown. (D) Immunoblotting for the indicated acetylated proteins in ASA-KO cells cultured in 10% or 1% FBS containing media with or

without 20-mM acetate for the indicated times. Ac histones were detected using an anti-acetyl-lysine motif antibody. (E) Immunoblotting for the designated acetylated and total protein levels in ASA-KO cells cultured in 10% or 1% FBS containing media with or without acetate for 4 hours. (F) Volcano plots for mRNA expression changes 4 hours after acetate removal in ASA-KO cells cultured in 10% or 1% FBS containing media. Down-regulated transcripts {$\log_2$ FC [(−) Acetate / (+) Acetate] < −1.0. q < 0.05}, and up-regulated transcripts {$\log_2$ FC [(−) Acetate / (+) Acetate] >1.0. q < 0.05} are shown in pink. Data shown represent the mean of 3 biological replicates. Values are from S2 Table (genes with q < 0.05 are shown). (G) GO analysis for down-regulated transcripts and up-regulated transcripts in the RNA sequencing with the 1% FBS condition. Enriched representative biological processes are shown. The data underlying the graphs in Fig 2 can be found in S1 Data. Ac, acetylated; acetyl-CoA, acetyl-coenzyme A; ASA-KO, acetate-supplemented ACLY knockout; DGLA, Dihomo-γ-linolenic acid; FFBS, fetal bovine serum; FC, fold change; GO, gene ontology; mRNA, messenger RNA; rRNA, ribosomal RNA.

transcripts in 10% FBS and 1% FBS conditions, respectively, $\log_2$ [FC: (−) Acetate / (+) Acetate] < −1.0, q < 0.05} (**Fig 2F** and **S2 Table**, sheet 1 and sheet 2 for 10% FBS and 1% FBS conditions, respectively). Intriguingly, we also found many transcripts that were significantly up-regulated in these conditions {365 and 899 transcripts in 10% FBS and 1% FBS conditions, respectively, $\log_2$ [FC: (−) Acetate / (+) Acetate] > 1.0, q < 0.05}. These differentially expressed transcripts in the 10% and 1% FBS conditions highly overlapped (70.5% or 33.8% of down-regulated transcripts in 10% or 1% FBS conditions, respectively, and 65.2% or 26.4% of up-regulated transcripts in 10% or 1% FBS conditions, respectively, **S3B Fig**), suggesting that a similar but stronger transcriptional response occurs in the 1% FBS condition. Gene ontology analyses for differentially expressed genes in the 1% FBS condition uncovered that several clusters of genes were similarly regulated (**Fig 2G**). For example, genes related to transcriptional regulation were highly enriched in both up- and down-regulated transcripts. Of particular interest was that genes involved in cellular stress responses such as apoptotic process, cell cycle arrest, and oxidation–reduction process were selectively up-regulated in acetate-withdrawn cells, suggesting that stress signaling pathways are activated in response to acetyl-CoA depletion.

## Acetyl-CoA depletion alters the integrity of the nucleolus

To further understand the complete cellular response after acetyl-CoA depletion, we also performed quantitative proteomics analyses using a tandem mass tag (TMT) system with the same experimental conditions used for our previous experiments (i.e., analysis 4 hours after acetate removal in 1% FBS containing media). Although overall alterations in protein levels appeared to be less drastic, we observed that the expression of 51 proteins were significantly increased more than 1.2-fold and 135 proteins were decreased less than 0.8-fold in response to acetate removal (**S3 Table**). Interestingly, we found that many proteins that are known to be localized to the nucleolus were significantly altered (**Fig 3A**, shown in red). Changes in these nucleolar proteins were not evident on the RNA sequencing analysis (**Fig 3A**, y-axis and **S2 Table**), implying that these proteins were likely regulated at a post-transcriptional level. By immunoblotting, we confirmed that the ribosome biogenesis factors RRP1 and BOP1 increased, while PICT1 (also known as NOP53 or GLTSCR2) decreased after acetate removal in ASA-KO cells but not in ASA-WT cells (**S4A and S4B Fig**). These observations indicate that alterations of the nucleolus might occur upon acetyl-CoA depletion. Hence, we next addressed whether acetyl-CoA depletion affected nucleolar structures and functions. The nucleolus is the organelle primarily responsible for ribosome biosynthesis, where rRNA synthesis, rRNA processing, and assembly of rRNA and ribosomal proteins take place [41,42]. Using a 5-Fluorouridine (FUrd) incorporation assay, we monitored newly synthesized rRNA in the nucleolus and found that acetate removal rapidly reduced rRNA synthesis in ASA-KO but not ASA-WT cells (**Fig 3B and 3C**). We also utilized an rRNA-specific dye to stain cells in conjunction with immunostaining for nucleolar proteins. In the presence of acetate, the nucleolar transcription factor upstream binding factor (UBF) localized within the nucleolar region as evident by its overlap with rRNA (**Fig 3D**). Consistent with the observed impairment of

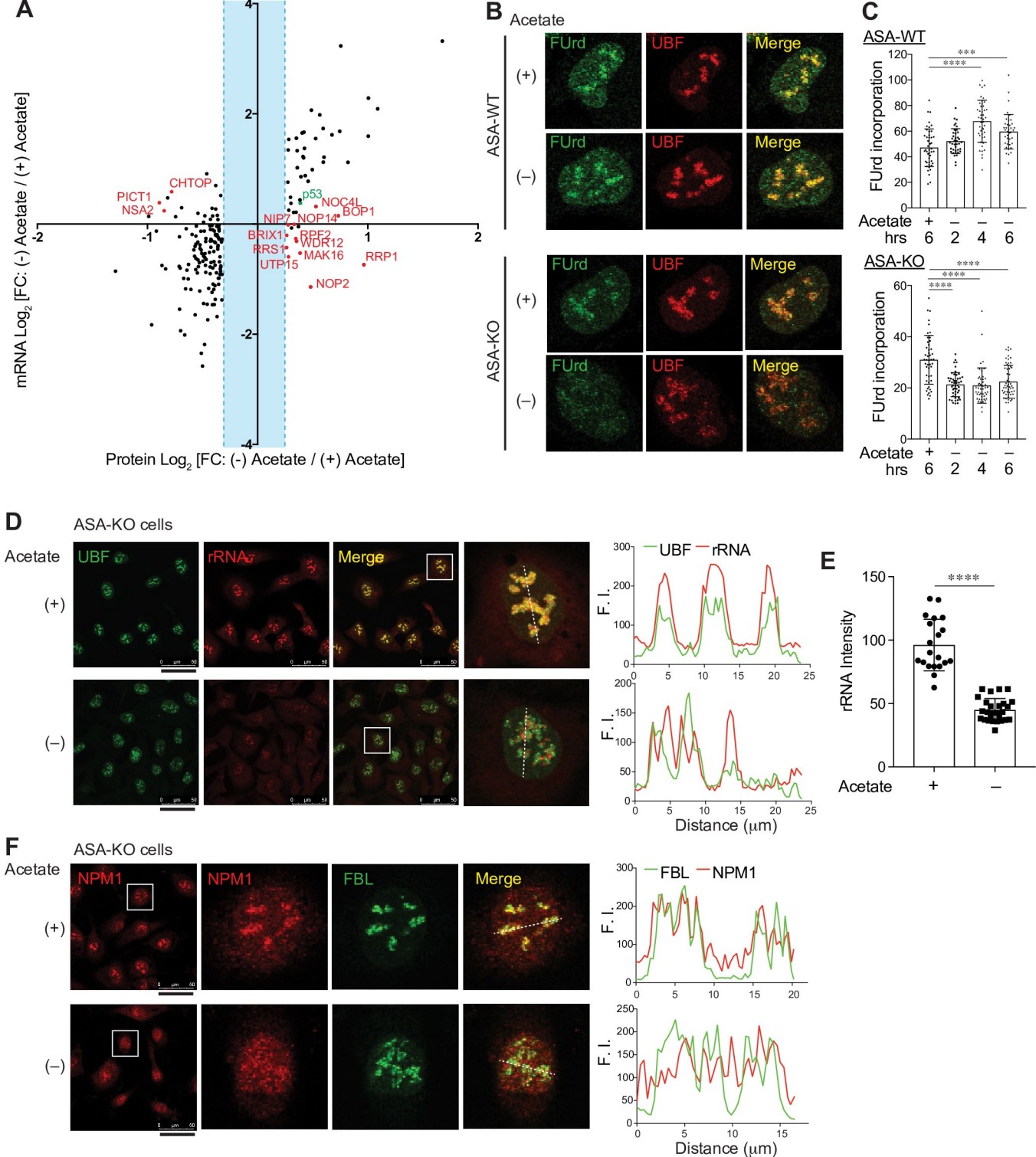

**Fig 3. Functional and structural remodeling of the nucleolus by acetyl-CoA depletion.** (A) Scatter plot for log2 FC in proteins (x-axis) and mRNAs (y-axis) 4 hours after acetate removal in ASA-KO cells. Values are from S2 and S3 Tables. Proteins with log2 FC more than 0.264 or less than −0.322 are shown. Proteins that are known to localize in the nucleolus are highlighted in red and p53 is shown in green. (B) FUrd incorporation assay in ASA-WT and ASA-KO cells cultured in 1% FBS containing media with or without 20-mM acetate over 6 hours. Representative nuclear images immunostained for FUrd and UBF are shown. (C) Quantification of the mean fluorescent intensity of the FUrd signal per nucleus. The microscopic images of the FUrd incorporation assay performed as in "B" were utilized for the quantification. Data are shown as mean ± SD (n = 39 to 55 cells per condition). ***P < 0.001, ****P < 0.0001 (1-way

ANOVA followed by Tukey multiple comparisons test). (D) Immunostaining for UBF along with rRNA dye staining in ASA-KO cells cultured in 1% FBS containing media with or without acetate for 4 hours. The scale bars under the left images indicate 50 μm. Magnified nuclear images (surrounded by a white square) are shown. Line profiles for indicated FI determined along the white dashed lines are shown to the right. (E) Quantification of the mean fluorescent intensity of the rRNA signal per nucleus in (D). Data are shown as mean ± SD [$n$ = 20 or 27 cells for (+) Acetate or (−) Acetate, respectively]. ****$P <$ 0.0001 (Unpaired $t$ test). (F) Immunostaining for NPM1 and FBL in ASA-KO cells cultured in 1% FBS containing media with or without acetate for 4 hours. Magnified nuclear images (surrounded by a white square) are shown. Line profiles for indicated FI determined along the white dashed lines are shown to the right. The data underlying the graphs in Fig 3 can be found in S1 Data. acetyl-CoA, acetyl-coenzyme A; ASA-KO, acetate-supplemented ACLY knockout; ASA-WT, acetate-supplemented ACLY wild type; FBL, fibrillarin; FBS, fetal bovine serum; FC, fold change; FI, fluorescent intensities; FUrd, 5-Fluorouridine; mRNA, messenger RNA; rRNA, ribosomal RNA; UBF, upstream binding factor.

rRNA synthesis (Fig 3B and 3C), fluorescent intensities for nuclear rRNA were significantly decreased in ASA-KO cells upon acetate removal (Fig 3D and 3E). Interestingly, residual rRNA signals detected in the acetate-deprived ASA-KO cells seemingly segregated from UBF (Fig 3D, magnified nuclear images with the line profiles). The nucleolar structure can be divided into 3 distinct functional subcompartments including the UBF containing compartment for rRNA synthesis (the fibrillar center), as well as compartments for rRNA splicing (the dense fibrillar component) and for rRNA-ribosomal protein assembly (the granular component) (S4C Fig) [43]. We therefore performed co-staining of rRNA with marker proteins for the latter 2 compartments, fibrillarin (FBL) or nucleolin (NCL), respectively. Interestingly, while both proteins colocalized with rRNA in the acetate-supplemented cells, neither FBL nor NCL fully merged with the segregated rRNA signals in the acetate-deprived ASA-KO cells (S4D and S4E Fig). These observations imply that acetyl-CoA depletion redistributes the rRNA containing compartment within the nucleolus. Furthermore, we found that in ASA-KO cells, the nucleolar localization of a nucleolar scaffolding protein, nucleophosmin 1 (NPM1), became dispersed following acetate removal (Fig 3F). We also noted that ASA-WT cells did not exhibit any of these morphological alterations in the nucleolus in response to acetate removal (S4F and S4G Fig). Altogether, these observations suggest that acetyl-CoA depletion markedly influences nucleolar components including changes in levels and in localization, thereby structurally and functionally remodeling the nucleolus.

## Acetyl-CoA depletion induces ribosomal protein-dependent p53 activation

In addition to the nucleolar alterations, we observed an increase in the level of p53 protein in the quantitative proteomics of acetyl-CoA–depleted cells (Fig 3A, shown in green). We also found up-regulation of the p53 target genes, including *MDM2*, *CDKN1A*, *BBC3*, and *PLK3*, in our RNA sequencing analysis (S5A Fig and S2 Table). We therefore next demonstrated by immunoblotting that removing acetate from the media increased p53 expression in ASA-KO cells (Fig 4A). In contrast, no increase in p53 upon acetate removal was observed in ASA-WT cells, suggesting that the acetate-dependent p53 induction is correlated with the fall of acetyl-CoA and with the nucleolar remodeling observed only in ASA-KO cells. A number of studies have demonstrated that so-called "nucleolar stress," which is caused by aberrations in ribosome biogenesis, induces p53 stabilization in a ribosomal protein–dependent manner [44,45]. Mechanistically, the most well-studied ribosomal proteins in this context are RPL11 and RPL5 [46–49]. These components are incorporated into a large ribosome subunit by forming a complex with 5S rRNA in the unstressed nucleolus. Once the nucleolus is stressed by, for instance, the RNA polymerase I inhibitor actinomycin D (ActD), the 5S-rRNA-RPL11-RPL5 ribonucleoprotein complex interacts with and inhibits MDM2 (also known as HDM2), an E3 ubiquitin ligase for p53, thereby stabilizing p53 (Fig 4B) [46,47]. Moreover, it has been reported that PICT1 is degraded by nucleolar stressors and that depletion of PICT1 is sufficient for p53 induction in a ribosomal protein–dependent manner [50,51]. Since we observed that acetyl-

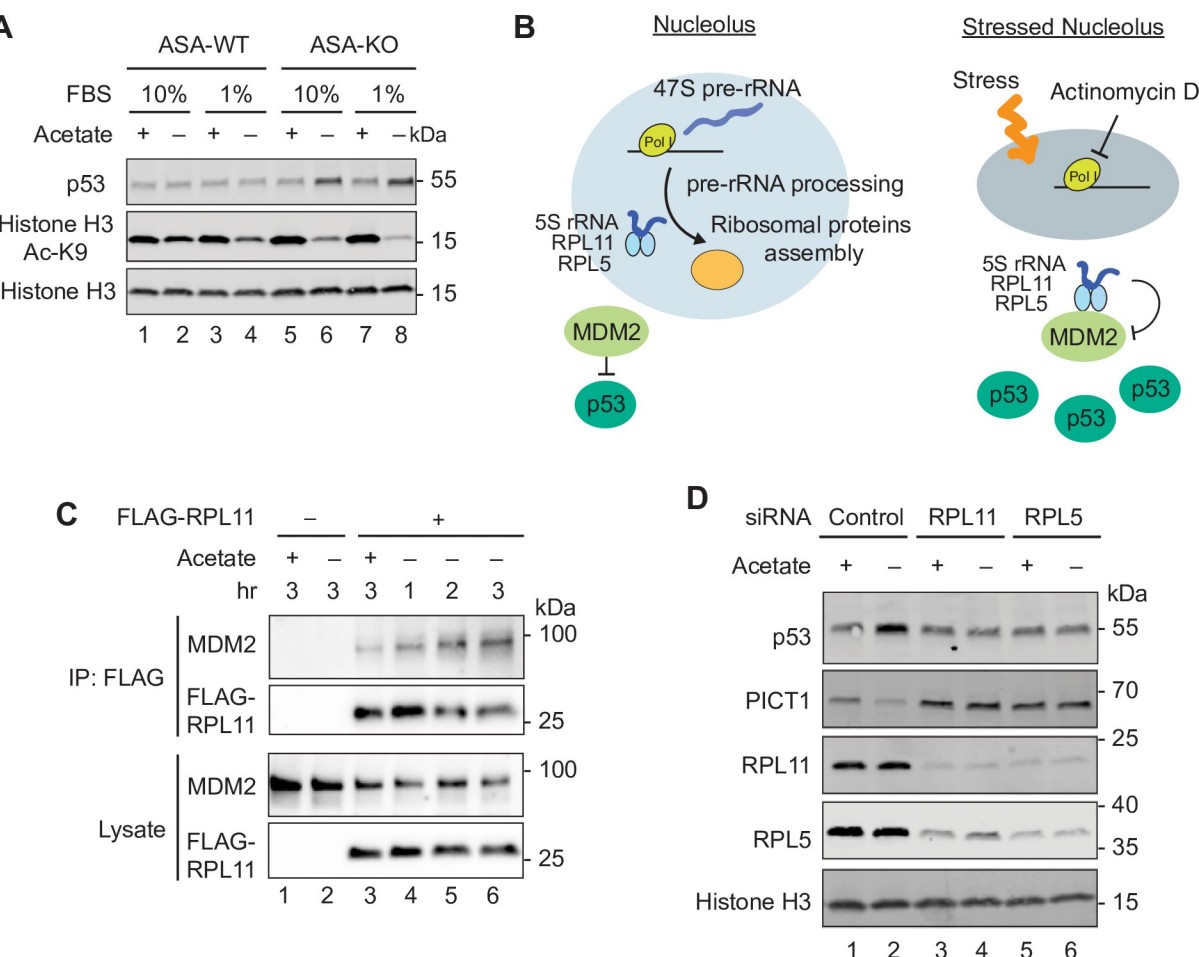

**Fig 4. Ribosomal protein–dependent p53 activation by acetyl-CoA depletion.** (A) Immunoblotting for p53 levels in ASA-WT and ASA-KO cells cultured in 10% or 1% FBS containing media with or without acetate for 4 hours. Ac K9 and total histone H3 are also shown. (B) Schematic representation of the nucleolus as an organelle for ribosome biogenesis (left) and the stressed nucleolus that induces p53 stabilization through the inhibitory interaction between the 5S rRNA-RPL11-RPL5 complex and MDM2. (C) Immunoprecipitation with FLAG-RPL11 and immunoblotting for MDM2 and FLAG-RPL11 in ASA-KO cells cultured in 1% FBS containing media with or without acetate for the indicated hours. (D) Immunoblotting for p53 levels in ASA-KO cells pretreated with indicated siRNAs and cultured in 1% FBS containing media with or without acetate for 4 hours. Histone H3 is shown as a loading control. Ac, acetylated; acetyl-CoA, acetyl-coenzyme A; ASA-KO, acetate-supplemented ACLY knockout; ASA-WT, acetate-supplemented ACLY wild type; FBS, fetal bovine serum; rRNA, ribosomal RNA; siRNA, small interfering RNA.

CoA depletion induced nucleolar alterations and reciprocal alterations in p53 and PICT1 levels, we next assessed whether these responses were mediated through a nucleolar stress condition. Upon acetate removal, interactions between endogenous MDM2 and FLAG-tagged RPL11 were found to increase in a time-dependent manner in ASA-KO cells (**Fig 4C**). Moreover, we tested small interfering RNA (siRNA)-mediated knockdown of RPL11 or RPL5 (**Fig 4D**). We note that knockdown of either RPL11 or RPL5 also reduced the expression of the other, implying an interdependency of their protein stability [48]. These knockdown cells had a reduced induction of p53 following acetate withdrawal, as well as reduced PICT1 degradation, suggesting that RPL11 and RPL5 are required for these processes (**Fig 4D**). These observations were consistent with other previous examples of nucleolar stress–dependent p53 activation [46,47]. DNA damage is a potent inducer of p53, but neither acetate removal nor ActD treatment induce phosphorylation of Serine 139 in Histone H2A.X (γH2A.X), a marker

for the DNA double strand breaks, suggesting DNA damage-independent mechanisms for the p53 up-regulation observed in acetyl-CoA–depleted cells (**S5B** and **S5C Fig**). Collectively, these observations suggest that acetyl-CoA depletion-induced p53 activation is mediated through ribosomal proteins associated with the nucleolar stress response.

Since the mechanistic target of rapamycin complex 1 (mTORC1) signaling has been demonstrated as a positive regulator of ribosome biosynthesis [52,53], and recent findings have indicated that mTORC1 activity is affected by acetyl-CoA levels [38,39], we examined whether acetyl-CoA depletion affected mTORC1 activity. Immunoblotting for phosphorylation of S6 kinase, a substrate of mTORC1, indicated that 4 hours of acetate removal, when the p53 induction was already observed in acetate withdrawn cells, did not reduce mTORC1 activity (**S5D Fig**). In contrast, Torin1, a chemical inhibitor of mTORC1, significantly suppressed mTORC1 signaling (**S5D Fig**). Also, unlike Torin1-treated cells that had a conversion of LC3 species from LC3-I to LC3-II, which indicated autophagy induction, we observed no significant alteration in LC3 upon acetate removal (**S5D Fig**). These results suggest that mTORC1 signaling and autophagy are not markedly affected following acute acetyl-CoA depletion in ASA-KO cells.

## Class IIa HDAC family mediates acetyl-CoA–dependent nucleolar alterations

We then addressed what mediates these nucleolus-dependent responses induced by acetyl-CoA depletion. As described above, we found drastic alterations in protein acetylation in acetyl-CoA–depleted ASA-KO cells (**Fig 2B–2E** and **S1 Table**). Therefore, we tested a possible role for lysine deacetylation in the acetyl-CoA depletion-dependent nucleolar alterations and subsequent p53 activation. The HDAC family members are the predominant deacetylases in mammalian cells, which can be grouped into zinc ion-dependent (class I, II, and IV) and NAD$^+$-dependent (class III, also known as Sirtuin family) families (**Fig 5A**) [54]. We utilized chemical inhibitors for HDACs as this enabled us to transiently inhibit HDACs following acetate depletion. We demonstrated that general inhibitors for zinc-dependent HDACs, Tricostatin A, and Vorinostat (also known as SAHA) completely inhibited the acetyl-CoA depletion-induced reciprocal regulation of p53 and PICT1 (**Fig 5B**), suggesting that zinc-dependent HDACs are likely required for this process. In contrast, EX-527, an inhibitor for the NAD$^+$-dependent deacetylase Sirtuin 1, did not alter this response (**S6A Fig**). Using selective inhibitors for each HDAC class, we sought to further address which HDAC class is responsible for the acetyl-CoA depletion-induced p53 and PICT1 regulation (**Fig 5A**). While the class I inhibitor Entinostat (also known as MS-275) [55] and the class IIa inhibitor TMP195 [56] exhibited differential inhibitory effect on deacetylation of histone H3 K9 and αTubulin K40, we found that TMP195 but not Entinostat blocked the reciprocal regulation of p53 and PICT1 (**Fig 5B**). TMP195 also suppressed the acetate-dependent induction of p21 (also known as CDKN1A), a downstream target of p53 (**Fig 5C**). Another class IIa inhibitor LMK235 [57] also exhibited the same effect, while other class I inhibitors including Mocetinostat [58] and RGFP966 (HDAC3 selective) [59] or the class IIb HDAC6 inhibitor Tubastatin A [60] did not (**Fig 5C** and **S6B Fig**). These observations let us hypothesize that the class IIa HDACs-mediated deacetylation may modulate the nucleolus. Thus, we investigated whether the inhibition of class IIa HDACs also maintained nucleolar integrity and rRNA synthesis using immunostaining with the rRNA dye and the FUrd assay, respectively. We demonstrated that the class IIa HDAC inhibitors TMP195 and LMK235 both restored the intensity of rRNA and its colocalization with NCL in acetyl-CoA–depleted ASA-KO cells (**S6C** and **S6D Fig**). Furthermore, TMP195 and LMK235 recovered FUrd incorporation while the class I inhibitor Entinostat was ineffective (**Fig 5D**

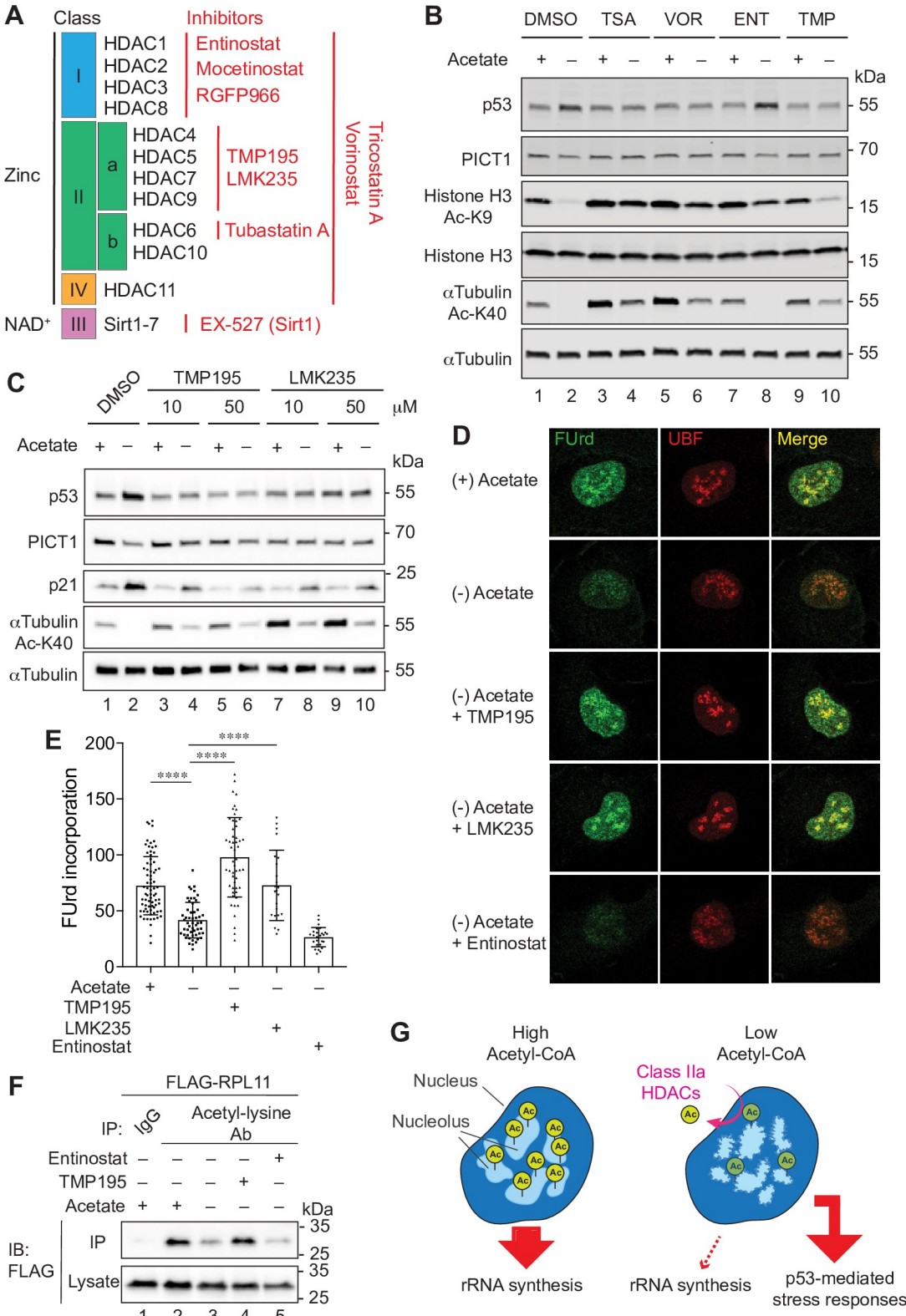

**Fig 5. Class IIa HDAC activity is required for acetyl-CoA–induced nucleolar stress responses.** (A) Schematic representation of mammalian HDAC family members and their inhibitors (red). (B) Immunoblotting for the indicated proteins in ASA-KO cells cultured in 1% FBS containing media with or without acetate, and in the presence or absence of indicated HDAC inhibitors for 4 hours. Following concentrations of HDAC inhibitors were used: 500-nM TSA, 10-μM VOR, 70-μM ENT, and

50-μM TMP. (C) Immunoblotting for the indicated proteins in ASA-KO cells cultured in 1% FBS containing media with or without acetate, and in the presence or absence of indicated class IIa HDAC inhibitors for 4 hours. (D) FUrd incorporation assay in ASA-KO cells cultured in 1% FBS containing media with or without acetate, and in the presence or absence of indicated HDAC inhibitors for 4 hours. Representative nuclear images (surrounded by a white square) immunostained for FUrd and UBF are shown. (E) Quantification of the mean fluorescent intensity of the FUrd signal per nucleus in "D." Data are shown as mean ± SD ($n$ = 23 to 74 cells per condition). ****$P$ < 0.0001 (1-way ANOVA followed by Tukey multiple comparisons test). (F) Immunoprecipitation with the acetyl-lysine motif antibody and immunoblotting for FLAG-RPL11 in ASA-KO cells cultured in 1% FBS containing media with or without acetate, and in the presence or absence of 50-μM TMP or 50-μM ENT for 90 minutes. (G) Model for the nucleolar responses upon nucleo-cytoplasmic acetyl-CoA fluctuations. See the main text for detail. The data underlying the graphs in Fig 5 can be found in S1 Data. Ab, antibody; acetyl-CoA, acetyl-coenzyme A; ASA-KO, acetate-supplemented ACLY knockout; ENT, Entinostat; FBS, fetal bovine serum; FUrd, 5-Fluorouridine; HDAC, histone deacetylase; IB, immunoblotting; IgG, immunoglobulin G; IP, immunoprecipitation; NAD, nicotinamide adenine dinucleotide; rRNA, ribosomal RNA; TMP, TMP195; TSA, Triconstatin A; UBF, upstream binding factor; VOR; Vorinostat.

and 5E). These findings suggest that the class IIa HDACs are selectively required for acetyl-CoA depletion-induced nucleolar remodeling and p53 activation.

## Class IIa HDACs regulate deacetylation of nucleolar proteins and nucleolar dynamics

To investigate whether the effect of class IIa HDAC inhibitors on the nucleolus is mediated through inhibition of deacetylation, we conducted acetylome analyses for acetyl-CoA–depleted cells with and without the class IIa inhibitor TMP195 and identified 365 acetylated peptides whose acetylation levels were maintained at a level 2-fold or greater by TMP195 treatment in the setting of acetate removal {S1 Table, column C, $\log_2$ [FC: (−) Acetate + TMP195 / (−) Acetate] >0.96}. These included proteins involved in cell–cell adhesion, mRNA splicing, rRNA processing, translational initiation, nucleosome assembly, and regulation of transcription (S7A Fig). Importantly, among these TMP195-sensitive proteins, we identified various nucleolus-resident proteins (S7B Fig). Among these, we first focused on acetylation of RPL11 (K8 and K67), since it is a critical regulator of the nucleolar stress-dependent p53 induction. Using the acetyl-lysine motif antibody immunoprecipitation assay, we demonstrated that the overall acetylation of RPL11 decreased upon acetate removal and that this was significantly suppressed by TMP195 but not by the class I inhibitor Entinostat (Fig 5F). These results suggest that deacetylation of RPL11 is selectively regulated by class IIa HDACs. We also examined deacetylation of NCL, as we detected TMP195-sensitive clustered acetylation (K70, K79, K87, K95, K102, K109, K110, K116, K124, and K132) in the N-terminal region of NCL, as well as several acetylation sites that were less sensitive to TMP195, which localized predominantly in the protein's carboxyl-terminal region (S7B and S7C Fig and S1 Table). Deacetylation of NCL by acetate removal was partially recovered by TMP195 (S7C Fig), suggesting that both class IIa and the other classes of HDACs may be involved in this regulation. These results suggest that the class IIa HDACs mediate deacetylation of at least certain nucleolar proteins including RPL11 and NCL.

Next, we sought to address an effect of acetylation/deacetylation in regulating the modified proteins. To examine whether acetylation affects molecular dynamics of the nucleolar proteins, we measured the mobility of NCL by using fluorescence recovery after photobleaching (FRAP) of monomeric GFP-tagged NCL (NCL-mGFP). We set a photobleaching area within the nucleolar region marked by NCL-mGFP so that we can monitor its mobility within the nucleolus. This experiment revealed that 90 minutes of acetate removal, when we observed marked deacetylation (Fig 2B–2D), there was also a modest but significant acceleration of the half recovery time of NCL-mGFP (S7D and S7E Fig). This result suggests that the mobility of NCL-mGFP within the nucleolus was facilitated by acetyl-CoA depletion, at least at this early

time point. Moreover, the presence of the class IIa inhibitor LMK235 but not the class I inhibitor Entinostat restored the half recovery times in acetate-depleted cells to similar levels as in acetate-supplemented cells (**S7D** and **S7E Fig**). We note that when compared with LMK235, TMP195 exhibited only partial effects on the FRAP of NCL-mGFP. This suggests that LMK235 and TMP195 may have slightly different selectivity for the target HDACs and that NCL may be regulated by multiple HDACs, as was also seen in the effect on acetylation of NCL (**S7C Fig**). Overall, these observations are consistent with a model in which class IIa HDACs-mediated deacetylation may alter the dynamics of nucleolar proteins.

## Acetylation-dependent regulation of the nucleolus in HCT116 cells

To extend our observations to additional cell types, we generated an HCT116 human colorectal carcinoma cell line–based ASA-KO cell line using the same CRISPR-mediated strategy as for HT1080 cells (**S1A Fig**). Again, viable *ACLY* KO cells were only achieved by supplementation of acetate in the culture media (**S8A Fig**). Acetate removal up-regulated p53 in HCT116 ASA-KO but not ASA-WT cells, and this was inhibited by the class IIa HDAC inhibitors (**S8A** and **S8B Fig**). Our HCT116 cells harbor yellow fluorescent protein (YFP)-tagged p53 (p53-YFP) stably expressing at low levels under non-stress conditions [61]. Upon acetate withdrawal, p53-YFP expression was similarly induced as endogenous p53, adding further evidence for the post-transcriptional regulation of p53 by acetyl-CoA depletion (**S8A** and **S8B Fig**). Moreover, class IIa HDAC-dependent nucleolar remodeling was also observed in HCT116 ASA-KO cells (**S8C Fig**). These observations extend the generality of our findings regarding acetylation-mediated regulation of the nucleolar stress responses.

## Discussion

In this study, we demonstrated that the nucleolus rapidly responds to acetyl-CoA depletion and activates stress response programs. We propose a model for this cellular response to a decline in acetyl-CoA nucleo-cytoplasmic levels (**Fig 5G**). In cells that contain high levels of acetyl-CoA, multiple nucleolar proteins are highly acetylated, and these modifications play a crucial role in maintaining the nucleolar integrity allowing for efficient rRNA synthesis. Once acetyl-CoA levels fall, class IIa HDACs deacetylate a significant number of nucleolar proteins. These reactions result in remodeling of the functional status of the nucleolus including the impairment of rRNA synthesis and the induction of the p53-mediated stress responses. This model highlights an important role of the nucleolus as an organelle sensor for coupling acetyl-CoA fluctuations to cellular stress responses. mRNA translation is known to be among the most energy-demanding processes in mammalian cells, and efficient ribosome biogenesis is necessary for cell growth and proliferation [62]. Therefore, it is likely that a fall in acetyl-CoA levels signals a state of energetic stress and that cells acutely reduce ribosome biogenesis in the nucleolus in order to limit energy consumption. Similar couplings of translational control and stress responses, but at different translational steps, are also observed in the integrated stress response [63,64].

Accumulating evidence has illustrated that the nucleolus serves as a stress responsive organelle toward a variety of stresses including genotoxic stress, oxidative stress, heat shock, nutrient deprivations, virus infections, and oncogene activations [44,45,65]. These nucleolar stresses, as well as chemical or genetic interventions of ribosome biogenesis, are known to induce morphological and functional alterations in the nucleolus, which are coupled with the activation of p53 or other signaling pathways in a stimulus- and cell context-dependent manner [44,66]. This study uncovered that the metabolite acetyl-CoA can be a novel modulator of the nucleolus through protein acetylation. It has been reported that rRNA synthesis and rRNA

processing are regulated by multiple mechanisms through nutrient signaling pathways to meet the cellular energy demands [52,53]. Together with these regulations, the cell likely uses this acetylation-mediated nucleolar stress response in order to help cope with physiological or pathological fluctuations in the cellular nutrient status.

We established the ASA-KO cell system as a cell model to selectively regulate nucleo-cytoplasmic acetyl-CoA levels through the acetate-ACSS2 pathway. Previous studies have reported other cell culture models to induce an acetyl-CoA decline in mammalian cells by modulating the citrate-ACLY pathway. Lee and colleagues demonstrated that culturing LN229 cells in the media containing 10-mM or 1-mM glucose, a major source of acetyl-CoA through the citrate-ACLY pathway, for 24 hours significantly decreased acetyl-CoA and increased CoA in the low glucose condition [29]. Furthermore, genetic deletion of *ACLY* in cultured mouse adipocytes exhibited similar but milder fluctuations in acetyl-CoA and CoA levels [30]. These models appear to require relatively longer time courses (24 hours or several days) to see a significant decrease in acetyl-CoA levels, which is probably because the acetate-ACSS2 pathway remains intact in these cells. Acetate-withdrawn ASA-KO cells can induce acute acetyl-CoA depletion (within 4 hours after acetate removal) since this condition blocks both the acetyl-CoA producing pathways. Hence, the ASA-KO system is suitable to investigate cellular responses to acute acetyl-CoA depletion.

Our findings demonstrated that class IIa HDACs-mediated deacetylation appears to be crucial for the acetyl-CoA depletion-induced nucleolar stress response. While the class IIa HDAC inhibitor TMP195 suppressed acetate-dependent p53 up-regulation, it had only minor effect on histone acetylation when compared with class I HDAC inhibitors such as Entinostat and Mocetinostat (**Fig 5B** and **S6B Fig**). These observations suggest that deacetylation of non-histone proteins by class IIa HDACs might be important in this context. Our acetylome analyses identified multiple acetylated lysine residues in various nucleolar proteins whose acetylation were increased in the presence of TMP195 (**S1 Table** and **S7B Fig**). Diversity of this list made it difficult to point out particular acetylation sites that are critical for the nucleolar integrity. We demonstrated that deacetylation of RPL11 appeared to be highly class IIa HDAC dependent (**Fig 5F**). Although the effect of this RPL11 acetylation is still undetermined, it might affect the nucleolar localization or protein stability of RPL11 as previously reported for other post-translational modifications (PTMs), such as Neddylation of RPL11 [67].

Recent quantitative proteomics studies validating the stoichiometry of acetylation at thousands of sites in HeLa cells have uncovered that most acetylation occurs at very low stoichiometry [68]. Although we have not determined the stoichiometry of acetylation in our ASA-KO cells, according to their data, stoichiometries of acetylation listed in **S7B Fig** are mostly around 0.05% to 0.5%. These observations suggest that local (i.e., around the nucleolus) or temporal regulation (i.e., upon acetate withdrawal) of acetylation may be crucial to modulate nucleolar integrity and the nucleolar stress responses. Also, it is likely that rather than acetylation of a single lysine residue or a single protein, the entire balance of nucleolar protein acetylation may determine a functional state of the nucleolus. In this regard, we do not exclude a possible involvement of acetylation of non-nucleolar proteins that reside in the environment surrounding the nucleolus, including the nuclear lamina, actin cytoskeleton, or chromatin, as these components are also known to influence the nucleolar structures and functions [69–73].

The nucleolus is known to be formed through a process called liquid–liquid phase separation (LLPS), a central mechanism for facilitating the formation of biomolecular condensates that exhibit liquid droplet-like or other related material properties [74–76]. Because LLPS is mediated through multivalent weak interactions among the component proteins and nucleotides, PTMs of proteins including phosphorylation, methylation, and acetylation can impact the assembly and material properties of the condensates by modulating their interactions

[76,77]. Rapid and reversible alterations of PTMs that are induced by various stimuli are capable of tuning the state of phase separation in a context-dependent manner. For instance, acetylation has been shown to regulate the LLPS of chromatin, as well as that of pathogenic protein aggregates, which are involved in neurodegenerative disorders, including Tau and TDP-43 [78–81]. It has been recently reported that LLPS-mediated formation of stress granules (SGs), membraneless organelles forming in response to stress, is modulated by acetylation of its component RNA helicase DDX3X [82]. In this case, deacetylation of DDX3X by HDAC6 is required for SGs maturation. Hence, it is tempting to speculate that the acetylation status of the nucleolar components may also participate in regulating the nucleolar phase separation and thereby affect its functional state. Our observations that acetyl-CoA depletion triggered redistribution of rRNA and NPM1, known contributors to the LLPS of the nucleolus (**Fig 3D–3F** and **S4D and S4E Fig**), implicate changes in the phase state of the nucleolus [83–85]. In addition, it appears that molecular dynamics of NCL is regulated by HDACs upon acetyl-CoA depletion (**S7D and S7E Fig**). As the molecular dynamics in a condensate can be seen as an indicator for its material property, these observations imply alterations in the phase material property of the nucleolus, which is reported to be tightly linked to the nucleolar function [86]. It would be worth noting that the clustered lysine residues that we found as TMP195-sensitive acetylation sites were located in the N-terminal intrinsically disordered region (IDR) of NCL (**S7C Fig**) [87]. It has been reported that, in general, IDRs play a central role for biomolecular LLPS and that PTMs in IDRs can influence the LLPS state [74–76]. Therefore, acetylation in the IDR of NCL, a nucleolar hub protein [87], might affect the phase separation state of the nucleolus by modulating the interaction mode of NCL. Further analyses will be required to conclude the effect of acetylation on the nucleolar LLPS.

Compared with the class I HDACs or the class IIb HDAC6, substrates of the class IIa HDACs have been less characterized [54,88]. It has been previously shown that the class IIa HDACs have only minimal deacetylase activity on acetylated histones in vitro [89,90]. Also, the transcriptional repressor activity by the class IIa HDACs expressed in cells or the enzymatic activity associated with immunoprecipitated class IIa HDACs are shown to be due to the deacetylase activity of the class I HDACs endogenously forming a complex with the class IIa HDACs [89,90]. A catalytic tyrosine residue within the HDAC domain conserved in all other HDACs was found to be replaced with histidine in the class IIa HDACs, which can explain their weak catalytic activity toward canonical HDAC substrates [89]. Therefore, the importance of the catalytic activity of the class IIa HDACs for their biological functions has been questioned. Our HDAC inhibitor profiling, however, demonstrated that the acetyl-CoA depletion-dependent nucleolar alterations and p53 induction was selectively sensitive to class IIa HDAC inhibitors, indicating that a class IIa HDAC catalytic domain-dependent reactions exist in this context. Our acetylome analyses with TMP195 provide a list of potential substrates for class IIa HDACs (**S1 Table**). This list highlights that various non-histone proteins including nucleolar proteins and ribosomal proteins are potential targets for this family of enzymes. Comparison with acetylome analyses using other class-selective HDAC inhibitors and in vitro reconstitution study will define bona fide substrates for the class IIa HDACs. In this study, we could not identify the specific HDAC most responsible for the acetyl-CoA–dependent nucleolar stress response among 4 class IIa HDAC family members. Given that all class IIa HDACs have similar secondary structures and that there appears to be a high degree of functional redundancy between the members [54], multiple HDACs might be involved in deacetylation of nucleolar proteins upon acetyl-CoA depletion. Future study will therefore be required to further delineate our initial observations.

In conclusion, we demonstrated that our ASA-KO cell system can be a useful tool for analyzing acute responses to acetyl-CoA depletion. Our findings indicate that acetylation-

mediated regulation of the nucleolus serves as an important hub linking acetyl-CoA metabolism to cellular stress responses. Furthermore, our findings highlight a potential application of the class IIa HDAC inhibitors for modulating nucleolar functions, which may provide novel insights for therapeutic interventions in a range of human diseases and in normal aging, where alterations or malfunctions of the nucleolus have been increasingly identified to play a role [91,92].

# Materials and methods

## Cell culture

HT1080 human fibrosarcoma cells (ATCC, CCL-121) were cultured at 37˚C with 5% $CO_2$ in Minimum Essential Medium (MEM) Eagle (with Earle's salts, L-glutamine, and sodium bicarbonate) (Sigma-Aldrich, St. Louis, Missouri), supplemented with 10% heat-inactivated FBS (Thermo Fisher Scientific, Asheville, North Carolina, Gibco or VWR Life Science, Radnor, Pennsylvania), 1x Penicillin Streptomycin Solution (Corning Incorporated, Corning, New York), and 1x MEM Nonessential Amino Acid Solution (Sigma). Sodium acetate (Sigma) was supplemented in the culture medium for the *ACLY* sgRNA-targeted cells. Dialyzed FBS was purchased from Thermo Fisher Scientific (Gibco). 293T cells (Lenti-X, Takara, San Francisco, California) and HCT116 cells [61] were cultured in the same condition as HT1080 cells but in Dulbeccos's Modified Eagle Medium (Gibco) supplemented with 10% FBS, 1-mM sodium pyruvate (Gibco), nonessential amino acids (Gibco), and GlutaMAX (Gibco).

## Plasmid constructions and lentivirus productions

Oligo duplexes containing an sgRNA sequence for ACLY (as shown in S1A Fig) were inserted into pCas9(BB)-2A-GFP (a gift from Feng Zhang, Addgene #48138) following the published procedures [93] to generate genome-edited cells using the CRISPR-Cas9 system. For lentivirus-mediated gene expression, a coding sequence (CDS) of human ACLY was PCR amplified from a HeLa cDNA pool and inserted into the *EcoR*I-digested pLVX-puro vector (Takara, San Francisco, California) using the In-Fusion cloning system (Takara). A CDS of RPL11 was amplified from an HT1080 cDNA pool and inserted into the *EcoR*I/*Bgl*II-digested p3xFLAG-CMV-7.1 (Sigma) to attach the 3xFLAG tag at its N-terminus. The 3xFLAG-RPL11 sequence was then inserted into the *EcoR*I-digested pLVX-puro vector. The NCL CDS was amplified from the GFP-NCL plasmid (a gift form Michael Kastan, Addgene #28176) [94]. The NCL-3xFLAG fragment was amplified using primers that attach the 3xFLAG at NCL's carboxyl terminus and was inserted into the *EcoR*I-digested pLVX-puro vector. For the NCL-mGFP expressing construct, the NCL-CDS was amplified from the GFP-NCL plasmid, and monomeric GFP (mGFP) was amplified from the LAMP1-mGFP plasmid (a gift from Esteban Dell'Angelica, Addgene #34831) [95], and these fragments were inserted into the *EcoR*I-digested pLVX-puro vector. Lentivirus was produced in 293T cells by transfecting the pLVX-puro vectors, psPAX2 (a gift from Didier Trono, Addgene #12260), and pCMV-VSV-G (a gift from Bob Weinberg, Addgene #8454) [96].

## Generation of the *ACLY* genome-edited cell lines by CRISPR-Cas9

The pCas9(BB)-2A-GFP plasmids containing the sgRNA sequence targeting *ACLY* were transfected into HT1080 cells or HCT116 p53-YFP cells using Lipofectamine LTX (Thermo Fisher Scientific) or X-treamGENE (Sigma) according to the manufacturer's instruction. The next day, GFP positive cells were collected using the FACS Fusion sorter (BD Biosciences, San Jose, California) and cultured in 3 different medium conditions; standard culture media or the

same media with 2- or 20-mM sodium acetate as illustrated in S1A Fig. Cell clones expanded from each condition were screened by genotyping PCR (the forward primer; gtggctgaagagc-tatgtccag and the reverse primer; cctctgcttgtgcacatctgtc) combined with *Xho*I digestion (as illustrated in S1B Fig) and by immunoblotting.

## Immunoblotting

ASA cells were plated at a density of $8 \times 10^4$ cells per well in a 6-well plate and cultured in the 20-mM sodium acetate–supplemented culture medium (10% FBS). The next day, the cells were washed with PBS twice and 1% FBS culture medium once, and then cultured in the media containing 10% or 1% FBS with or without 20-mM sodium acetate, or with indicated chemicals according to experiments. After indicated time periods, the cells were washed with PBS or 1% FBS culture medium once and lysed in 150 ul of 1x NuPAGE LDS sample buffer (Thermo Fisher Scientific) supplemented with 10-mM DTT. The samples were shaken at max speed in a table top incubator at room temperature for 10 minutes, boiled at 98˚C for 5 minutes, and centrifuged at 15,000 rpm for 5 minutes. The supernatant containing approximately 15- to 30-μg protein was separated by SDS-PAGE using a 4% to 20% gradient Mini-PROTEAN TGX Precast Gel (Bio-Rad, Hercules, California) and then transferred to a 0.2-μM nitrocellulose membrane. The membrane was blocked with Odyssey Blocking Buffer (LI-COR, Lincoln, Nebraska) and incubated with the indicated primary antibodies at 4˚C overnight. After washing with PBS-T (PBS + 0.05% Tween-20), the membrane was incubated with near-infrared fluorescent IRDye secondary antibodies (LI-COR) or HRP-conjugated secondary antibodies (Thermo Fisher Scientific) and washed again with PBS-T. Detection was performed with either the ODYSSEY CLx Infrared Imaging System together with Image Studio Lite version 5.2 Software (LI-COR) or iBright CL1000 Imaging System (Thermo Fisher Scientific). For immunoblotting of BOP1 and RRP1, the cells were lysed with 1% Triton buffer [1% Triton-X100, 150-mM NaCl, 50-mM Tris-HCl (pH 7.4), 1-mM EDTA, Phosphatase inhibitors (PhosSTOP, Sigma), and protease inhibitors (cOmplete, Sigma)]. After centrifugation, the supernatant was mixed with 4x NuPAGE LDS sample buffer and boiled. For siRNA-mediated knockdown experiments, silencer siRNAs (5 nM) for indicated genes or Negative Control siRNA #1 (Ambion, AM4611) were transfected with Lipofectamin RNAi MAX transfection reagent (Thermo Fisher Scientific, 13778075) into $4 \times 10^4$ cells in a 6-well plate. The next day, the culture media were replaced with fresh media, and the day after that, the cells were subjected to immunoblotting experiments. Following Silencer Select, predesigned siRNAs were used; RPL5 (ID s56731), RPL11 (ID s12168).

## Immunoprecipitation

ASA-KO17 cells and the ASA-KO17 cells stably expressing 3xFLAG-RPL11 were plated at a density of $7 \times 10^5$ cells per 10-cm culture dish and cultured in the 20-mM sodium acetate–supplemented culture medium (10% FBS). The next day, the cells were washed with PBS twice and 1% FBS culture medium once, and then cultured in 1% FBS culture medium with or without 20-mM sodium acetate (two 10-cm dishes were prepared for each condition). After indicated time periods, the cells were washed with PBS and lysed in 500 μl of 1% Triton buffer [1% Triton-X100, 150-mM NaCl, 50-mM Tris-HCl (pH 7.4), 1-mM EDTA, Phosphatase inhibitors (PhosSTOP, Sigma), and protease inhibitors (cOmplete, Sigma)]. After centrifugation, 30 μl of the supernatant was taken for "Lysate" sample, and the rest of the supernatant was mixed with 10 μl of Anti-FLAG M2 Magnetic beads (Sigma). After 30 minutes of incubation at 4˚C, the beads were washed with 1% Triton buffer 3 times and then incubated in 30 μl of 1% Triton buffer with 0.2 μg/μl of 3xFLAG peptide (Sigma) at room temperature for 20 minutes. Then,

the supernatant was mixed with 4x NuPAGE LDS sample buffer (Thermo Fisher Scientific) supplemented with 10-mM DTT and boiled.

The immunoprecipitation assay was performed using an acetyl-lysine motif antibody to detect acetylated protein. The ASA-KO17 cells stably expressing 3xFLAG-RPL11 or NCL-3xFLAG were plated, stimulated, and lysed as the FLAG immunoprecipitation assay described above. The supernatant was mixed with either 0.8-μg Rabbit IgG (Santa Cruz, Dallas, Texas, sc-2027) or 4 μl of the acetyl-lysine motif antibody (Cell Signaling Technology, Danvers, Massachusetts, #9814) and incubated over night at 4˚C. Then, the supernatant was mixed with 10 μl of Protein A/G Magnetic Beads (Thermo Fisher Scientific) and incubated for 1 hour at 4˚C. The beads were washed with 1% Triton buffer 3 times and mixed with 35 μl of 1x NuPAGE LDS sample buffer supplemented with 10-mM DTT and shaken at room temperature for 10 minutes. Then, the supernatant was subjected to SDS-PAGE.

## Cell viability analysis (WST-1 assay)

Cells were plated at a density of 5,000 cells per well in a 6-well plate and cultured in the culture medium (10% FBS) with or without 20-mM sodium acetate. The next day, the cells were washed with PBS twice and 1% FBS culture medium once, and then cultured in the media containing 10% FBS, 1% FBS, or 10% dialyzed FBS with or without 20-mM sodium acetate. Twenty-four hours later, the culture media were replaced with 10% FBS culture medium with or without 20-mM sodium acetate, and the cells were cultured for another 2 days. Then, the media were replaced with 10% FBS medium containing 50-μM WST-1 (Dojindo, Gaithersburg, Maryland) and 20-μM 1-methoxy PMS (Dojindo), and the cells were incubated for 2 hours at 37˚C in the cell culture incubator before the absorbance at 440 nm in the culture media (100-μl aliquots transferred in a 96-well plate) was measured using Spectra Max Plus 384 microplate Reader (Molecular Devices, Sunnyvale, California).

## Quantification of acetyl-CoA and Coenzyme A by HPLC with UV detection

ASA-WT5 and ASA-KO17 cells were plated at a density of $4 \times 10^5$ cells per 10-cm culture dish and cultured in the 20-mM sodium acetate–supplemented culture medium (10% FBS) for 1 day (10 dishes were prepared for each cell line). The cells were then washed with PBS twice and cultured in 1% FBS culture media with or without 20-mM sodium acetate (5 dishes were prepared for each condition). After 4 hours, cells were washed with PBS, detached by trypsin, and resuspended in the culture media (approximately 1.5 to $2.0 \times 10^6$ cells). Cell pellet was collected by centrifugation, washed with PBS twice, and resuspended in 100 μl of 5% 5-Sulfosalicylic acid (Sigma) solution. To permeabilize the cells, samples were frozen in liquid nitrogen and thawed on ice. This freeze–thaw cycle was repeated twice. After centrifugation, the supernatant was filtered using Ultrafree-MC LH Centrifugal Filter (Millipore, Billerica, Massachusetts, UFC30LH25). Then, the samples were transferred into SUN-SRi Glass Microsampling Vials (Thermo Fisher Scientific, 14-823-359) with SUN-SRi 11mm Snap Caps (Thermo Fisher Scientific, 14-823-379), and 80 μl of each sample was separated using an Agilent 1100 HPLC (Agilent Technologies, Santa Clara, California) equipped with a reverse phase column, Luna 3 μm C18(2) 100 Å, 50 × 4.6mm, 3 μm (Phenomenex, Los Angeles, California). The HPLC-reverse phase column was calibrated with acetyl-CoA (Sigma, A2056) and coenzyme A (Sigma, C3144). The mobile phase was described previously [97] and consisted of 2 eluents: 75-mM sodium acetate and 100-mM $NaH_2PO_4$ (pH 4.6) (buffer A). Methanol was added to the buffer A at a ratio of 70:30 (v/v) = buffer A:methanol (buffer B). The gradient started with 10% of buffer B and was run under the following conditions: 10 minutes at up to 40% of buffer B, 13 minutes at up to 68% of buffer B, 23 minutes at up to 72% of buffer B, 28 minutes at up

to 100% of buffer B, and held for an additional 5 minutes. The initial condition was restored after 10 minutes with 10% of buffer B. The flow rate was 0.6 ml/min, and the detection was performed at 259 nm. UV chromatograms were analyzed using the software ChemStation version A.10.01 (Agilent Technologies). Standard curves of acetyl-CoA and CoA were prepared in the range of 10 to 1000 pmol and 100 to 400 pmol, respectively. To estimate absolute concentrations of acetyl-CoA and CoA in a single cell, cell sizes of trypsinized ASA-WT and ASA-KO were measured using Countess II FL Automated Cell Counter (Thermo Fisher Scientific), and cell volumes were determined assuming the cell shape as a sphere. Determined volumes were as follows: ASA-WT cells with acetate, 4.69 pL; ASA-WT cells without acetate, 5.25 pL; ASA-KO cells with acetate, 8.77 pL; and ASA-WT cells without acetate, 8.86 pL.

## Quantification of acetyl-CoA and Coenzyme A by high-resolution LC-MS

ASA-KO17 cells were plated at a density of $4 \times 10^5$ cells per 10-cm culture dish and cultured in the 20-mM sodium acetate–supplemented culture medium (10% FBS) for 1 day. The cells were then washed with PBS twice and 1% FBS culture media once and cultured in the 1% FBS culture media with or without 20-mM sodium acetate (2 dishes were prepared for each condition). After 4 hours, cells were washed with PBS twice and collected in 1 ml of 80% methanol (approximately 2 to $3 \times 10^6$ cells). Metabolic quenching and polar metabolite pool extraction was performed using ice cold 80% methanol / 0.1% formic acid at a ratio of 1 mL per 1 million adherent cells. Stable isotope–labeled ($^{13}C_2$) acetyl-CoA and ($^{13}C_1$)-creatinine (Sigma) was added to the sample lysates as an internal standard for a final concentration of 10 μM. After 3 minutes of vortexing, the supernatant was cleared of protein by centrifugation at 16,000 *xg*. A total of 3 μL of cleared supernatant was subjected to online LC-MS analysis. Standard stock solutions of CoA and acetyl-CoA (25 to 0.09 μM) were prepared in deionized water, and a standard curve was prepared by serial dilutions of the stock solution. Analyses were performed by untargeted LC-HRMS. Briefly, samples were injected via a Thermo Vanquish UHPLC and separated over a reversed phase Phenomenex Luna C18 [2] column (2.1 × 100 mm, 5-μm particle size) maintained at 55°C. For the 20-minute LC gradient, the mobile phase consisted of the following: solvent A (water / 5-mM ammonium acetate) and solvent B (ACN / 5-mM ammonium acetate). The gradient was the following: 0 to 1.5 min 2% B, increase to 98% B over 11 minutes, hold 98% B over 4 minutes, reequillibrated at 2% B for 4 minutes. The Thermo IDX tribrid mass spectrometer was operated in positive mode, scanning in Full MS mode (2 μscans) from 100 to 800 m/z at 70,000 resolution with an AGC target of 2e5. Heated electrospray ionization source (HESI) spray voltage was set to 3.0 kV. Source gas parameters were 35 sheath gas, 12 auxiliary gas at 320°C, and 8 sweep gas. Calibration was performed prior to analysis using the Pierce FlexMix Ion Calibration Solutions (Thermo Fisher Scientific). Integrated peak areas were then extracted manually using Quan Browser (Thermo Fisher Xcalibur ver. 2.7) and normalized to internal standard peak areas. Concentrations of acetyl-CoA and CoA in a single cell were determined as described in the HPLC-UV section.

## Quantification of cholesterol and fatty acids

ASA-KO17 cells were plated and stimulated, and the cell pellet (approximately 1.1 to $2.8 \times 10^6$ cells) was collected as described in the section for quantification of acetyl-CoA. For measurement of cholesterol and fatty acids, 500 μL of aqueous cell lysate was extracted with 2 mL of chloroform:methanol (2:1) after the addition of 10 μL of a deuterated fatty acid internal standard mix (50 ppm) and 20 μL of cholesterol-$d_7$ (10 ppm). The samples were vortexed and centrifuged at 2,500 rpm for 10 minutes at 4°C, and the organic layer was transferred into 2 new glass vials in equal volumes for fatty acid derivatization and cholesterol analysis. Both vials

were dried under $N_2$. Samples for cholesterol analysis were reconstituted in 100 μL of chloroform:methanol (2:1), and 10 μL was injected into a Shimadzu LC with CTC PAL autosampler coupled to a Sciex 5000 triple quadrupole mass analyzer. Analytes were separated on a C18(2) Luna column (2.1 × 100 mm, Phenomenex) at a flow rate of 0.63 mL/min using 50:50 $H_2O$: ACN with 0.1% formic acid for solvent A and 40:60 ACN:IPA with 0.1% formic acid for solvent B. The gradient started at 50% B and increased to 100% B over 6 minutes before returning to initial conditions for column equilibration. Cholesterol and cholesterol-$d_7$ were detected by selective reaction monitoring using 369→147 and 376→147 transitions, respectively. Peak area ratios of cholesterol to cholesterol-$d_7$ were normalized to cell number and are reported as relative amount. Fatty acids were derivatized for LC-MS analysis using a Vanquish UPLC coupled to a Q Exactive mass spectrometer (Thermo Fisher Scientific). Derivatization of the samples was conducted as described [98]. To each sample, 200 μL of oxalyl chloride (2 M in dichloromethane) was added, and the mixture was incubated for 5 minutes at 65˚C. Samples were dried under $N_2$, and 150 μL of 3-picolylamine (1% v/v in ACN) was added for a 5-minute incubation at room temperature to form the 3-picolylamide (PA) ester. Samples were dried a final time under $N_2$ and reconstituted in 1-mL MeOH, and 5 μL was injected onto the Vanquish-QE system. Fatty acid-PA esters were separated on a Phenomenex C8 column (2.1 × 150 mm, 5-μ pore size) using $H_2O$ + 0.1% acetic acid for solvent A and ACN + 0.1% acetic acid for solvent B. The gradient started at 65% B and increased linearly to 85% B at 10 minutes and was held for 1 minute before ramping to 100% B for 2 minutes. Finally, the gradient returned to 65% B for a 2-minute equilibration. Samples were analyzed using full scan accurate mass at a resolution of 70K. Peak area ratios of fatty acid-PA ester to corresponding internal standard were normalized to cell number and reported as relative amount.

## Acetylome analysis

ASA-KO17 cells were plated at a density of $6 \times 10^6$ cells per 15-cm culture dish and cultured in the 20-mM sodium acetate–supplemented culture media (10% FBS) for 1 day. The cells were then washed with PBS twice and 1% FBS culture medium once, and cultured in 1% FBS culture media with or without 20-mM sodium acetate, or without acetate but with 50-μM TMP195 (21 dishes were prepared for each condition). After 90 minutes, the cells were washed with 10 ml of cold PBS once and lysed in 10 ml of PTMScan Urea Lysis Buffer [20-mM HEPES (pH 8.0), 9.0-M urea, 1-mM sodium orthovanadate (activated), 2.5-mM sodium pyrophosphate, 1-mM β-glycerol-phosphate] (Cell Signaling Technology). Samples were frozen in dry ice/ethanol and then analyzed using the PTMScan method (Cell Signaling Technology) as previously described [99,100]. Briefly, the lysates were sonicated, centrifuged, reduced with DTT, and alkylated with iodoacetamide. A total of 15 mg of total protein for each sample was digested with trypsin and purified over C18 columns for enrichment with the Acetyl-Lysine Motif Antibody (#13416). Enriched peptides were purified over C18 STAGE tips. Enriched peptides were subjected to secondary digest with trypsin and second STAGE tip prior to LC-MS/MS analysis. Replicate injections of each sample were run nonsequentially on the instrument. Peptides were eluted using a 90-minute linear gradient of acetonitrile in 0.125% formic acid delivered at 280 nL/min. Tandem mass spectra were collected in a data-dependent manner with a Thermo Orbitrap Fusion Lumos Tribrid mass spectrometer using a top-20 MS/MS method, a dynamic repeat count of 1, and a repeat duration of 30 seconds. Real-time recalibration of mass error was performed using lock mass with a singly charged polysiloxane ion m/z = 371.101237. MS/MS spectra were evaluated using SEQUEST and the Core platform from Harvard University. Files were searched against the SwissProt *Homo sapiens* FASTA database. A mass accuracy of +/−5 ppm was used for precursor ions and 0.02 Da for product ions. Enzyme specificity was limited to trypsin, with

at least 1 tryptic (K- or R-containing) terminus required per peptide and up to 4 miscleavages allowed. Cysteine carboxamidomethylation was specified as a static modification; oxidation of methionine and acetylation on lysine residues were allowed as variable modifications. Reverse decoy databases were included for all searches to estimate false discovery rates and filtered using a 2.5% FDR in the Linear Discriminant module of Core. Peptides were also manually filtered using a −/+ 5ppm mass error range and presence of an acetylated lysine residue. All quantitative results were generated using Skyline to extract the integrated peak area of the corresponding peptide assignments. Accuracy of quantitative data was ensured by manual review in Skyline or in the ion chromatogram files. Gene ontology (GO) analysis was performed using functional annotation in DAVID Bioinformatics Resources (david.ncifcrf.gov).

### RNA sequencing analysis

ASA-KO17 cells were plated at a density of $4 \times 10^5$ cells per 10-cm culture dish and cultured in the 20-mM sodium acetate–supplemented FBS culture media (10% FBS) for 1 day. The cells were then washed with PBS twice and cultured in the culture media containing 10% or 1% FBS with or without 20-mM sodium acetate (2 dishes were prepared for each condition). After 4 hours, cells were washed with PBS, collected in 1-mM EDTA in PBS, and total RNAs were isolated using RNeasy Mini Kit (Qiagen, Germantown, Maryland, 74106) according to the manufacturer's instruction. The sequencing libraries were constructed from 100 ng to 500 ng of total RNA using the Illumina's TruSeq Stranded Total RNA kit with Ribo-Zero following the manufacturer's instruction. The fragment size of RNAseq libraries was verified using the Agilent 2100 Bioanalyzer (Agilent Technologies, Santa Clara, California), and the concentrations were determined using Qubit instrument (Thermo Fisher Scientific). The libraries were loaded onto the Illumina HiSeq 3000 for $2 \times 75$ bp paired-end read sequencing. The fastq files were generated using the bcl2fastq software for further data analysis. Data analysis: After quality assessment, adapter and low-quality bases trimming, reads were aligned to the reference genome using the latest version of HISAT2 [101], which sequentially aligns reads to the known transcriptome and genome using the splice-aware aligner built upon HISAT2. Only uniquely mapped paired-end reads were then used for subsequent analyses. FeatureCounts [102] was used for gene level abundance estimation. Differential expression analysis was then carried out using open source Limma R package [103]. Limma-voom was employed to implement a gene-wise linear modeling, which processes the read counts into $\log_2$ counts per million (logCPM) with associated precision weights. The logCPM values were normalized between samples using trimmed mean of M-values (TMM). We adjust for multiple testing by reporting the FDR q-values for each feature. Features with q < 5% were declared as genome-wide significant. GO analysis was performed using functional annotation in DAVID Bioinformatics Resources (david.ncifcrf.gov).

### Tandem mass tag mass spectrometry analysis

ASA-KO17 cells were plated at a density of $4 \times 10^5$ cells per 10-cm culture dish and cultured in the 20-mM sodium acetate–supplemented culture media (10% FBS) for 1 day. The cells were then washed with PBS twice and cultured in 1% FBS culture media with or without 20-mM sodium acetate (2 or 3 dishes were prepared for each condition). After 4 hours, cells were washed with ice-cold PBS 3 times, harvested with 1-mM EDTA in PBS, collected by centrifugation, and lysed with RIPA buffer [1% NP-40, 150-mM NaCl, 50-mM Tris-HCl (pH 7.4), 0.1% SDS, 0.5% Na-DOC, Phosphatase inhibitor (PhosSTOP, Sigma), and protease inhibitor (cOmplete, Sigma)]. After centrifugation, the supernatant containing 100-μg protein was used for the tandem mass spectrometry analysis. The samples were reduced, alkylated, digested, and labeled according to instructions for the TMT 10 plex kit (Thermo Fisher Scientific). TMT-

labeled peptides were mixed, desalted, and fractionated on an Agilent 1200 HPLC system to 24 fractions using basic reversed-phase chromatography. LC-MS/MS was performed on a Dionex UltiMate 3000 rapid separation nano UHPLC system (Thermo Fisher Scientific) coupled online to an Orbitrap Fusion Lumos tribrid mass spectrometer (Thermo Fisher Scientific). Peptides were first loaded onto a nano trap column (Acclaim PepMap100 C18, 3 μm, 100Å, 75 μm i.d. × 2 cm, Thermo Fisher Scientific), and then separated on a reversed-phase EASY--Spray analytical column (PepMap RSLC C18, 2 μm, 75 μm i.d. × 50 cm, Thermo Fisher Scientific) using a linear gradient of 4% to 32% B (buffer A: 0.1% formic acid in water; buffer B: 0.1% formic acid in acetonitrile) for 100 minutes. The mass spectrometer was equipped with a nano EASY-Spray ionization source, and eluted peptides were brought into gas-phase ions by electrospray ionization and analyzed in the orbitrap. High-resolution survey MS scans and HCD fragment MS/MS spectra were acquired in a data-dependent manner with a cycle time of 3 seconds. Dynamic exclusion was enabled. Raw data files generated from LC-MS/MS were analyzed using a Proteome Discoverer v2.2 software package (Thermo Fisher Scientific) and the Mascot search engine (Matrix Science, Boston, Massachusetts). The following database search criteria were set to: database, SwissProt human; enzyme, trypsin; max miscleavages, 2; variable modifications, oxidation (M), deamidation (NQ); fixed modifications, TMT (K, N-term), carbamidomethylation (C); peptide precursor mass tolerance, 20 ppm; and MS/MS fragment mass tolerance, 0.03 Da. Peptide-spectrum matches (PSMs) were filtered to achieve an estimated false discovery rate (FDR) of 1% based on a target–decoy database search strategy. The relative abundance of a protein or peptide in different samples was estimated by TMT reporter ion intensities.

## Light microscopy analysis

Cells were plated at a density of $4 \times 10^4$ cells per well in a 4-well chamber slide (Nunc Lab-Tek II Chamber slide system, Thermo Fisher Scientific) and cultured in the 20-mM sodium acetate–supplemented culture medium (10% FBS). The next day, the cells were washed with PBS twice and 1% FBS culture medium once, and cultured in 1% FBS culture media with or without 20-mM sodium acetate, or with or without indicated compounds. After indicated periods, the cells were washed with PBS, fixed with 4% formaldehyde/PBS for 10 minutes, permeabilized with 1% Triton-X/PBS for 15 minutes, and blocked with 2% BSA/PBS for 30 minutes at room temperature. Primary antibodies in 2% BSA/PBS were incubated overnight at 4˚C, and secondary antibodies (Thermo Fisher Scientific, Alexa Fluor) were incubated for 2 hours at room temperature and washed with PBS-T. Of note, 1-μM Nucleolus bright red (Dojindo) was treated in PBS for 10 minutes and washed with PBS just before the observation. For FUrd incorporation assay, 5-Fluorouridine (1 mM) (Sigma) was added in the culture media 15 minutes before fixation. The FUrd signal was detected by staining with anti-BrdU [BU1/75 (ICR1)] Rat mAb (abcam, Boston, Massachusetts, ab6326) and Alexa Fluor 488 anti-Rat IgG (Jackson Laboratory, Bar Harbor, Maine, 712-547-003). Fluorescent images were acquired using Leica SP8 LIGHTNING Confocal Microscope (Leica, Buffalo Grove, Illinois) equipped with a 63 × /1.4 NA oil immersion objective and driven by LAS X software. Image quantifications were performed with LAS X software. Fluorescent images in S5C Fig were acquired using a Zeiss LSM 780 Confocal Microscope (Carl Zeiss, Dublin, California) equipped with a 63 × /1.4 NA oil immersion objective and driven by ZEN software.

## FRAP analysis

ASA-KO17 cells were plated at a density of $2 \times 10^5$ cells per well in a 35-mm glass bottom dish (No. 1.5 Coverslip, 14-mm glass diameter, uncoated, MatTek) and cultured in the 20-mM

sodium acetate–supplemented culture medium (10% FBS). The next day, the cells were washed with PBS twice and 1% FBS culture medium once, and cultured in 1% FBS culture media with or without 20-mM sodium acetate, or with or without indicated compounds. A total of 90 minutes later, confocal images were taken on a Leica SP8 confocal microscope with a HC PL APO 1.30 NA 93x glycerin objective using a Acousto-Optic Beam Splitter (AOBS) emission system on LASX software (Leica, v3.5.2.18963). Images were acquired at 1.54 frames per second, with 132-nm pixels using a galvo scanner at 400 Hz. Marker mobility was measured using the FRAP module, bleaching within the ROI with 3 passes of high power 405-nm laser. Immediately post-bleach, samples were imaged continuously for approximately 50 seconds. T1/2 was measured using Elements (Nikon, Melville, New York, v5.210) time measurement tool.

## Reagents and antibodies

Reagents were sourced as indicated: Tricostatin A (Sigma, T1952), Vorinostat (SAHA, MK0683) (Selleck Chemicals, Houston, Texas, S1047), Entinostat (MS-275) (Selleck Chemicals, S1053), Mocetinostat (Cayman Chemical, Ann Arbor, Michigan, 18287), RGFP966 (Cayman Chemical, 16917), TMP195 (Cayman Chemical, 23242), LMK235 (Cayman Chemical, 14969), Tubastatin A (Apex Biomedical, Clackamas, Oregon, A4101), Camptothecin (Selleck Chemicals, S1288), Actinomycin D (Sigma, A1410), and Torin1 (Cayman Chemical, 10997). Antibodies were sourced as indicated: ATP-Citrate Lyase (Cell Signaling Technology, #4332), Histone H3 (Cell Signaling Technology, #4499), Acetyl-Histone H3 (Lys9) (Cell Signaling Technology, #9649), Acetyl-Histone H3 (Lys27) (Cell Signaling Technology, #8173), Acetyl-Histone H4 (Lys8) (Cell Signaling Technology, #2594), Phospho-Histone H2A.X (Ser139) (Cell Signaling Technology, #9718), Histone H2A (Cell Signaling Technology, #12349), Acetylated-Lysine Antibody (Cell Signaling Technology, #9814), α-Tubulin (1,5000, Sigma, T6199), Acetyl-α-Tubulin (Lys40) (Cell Signaling Technology, #5335), p53 (Santa Cruz, sc-126), GLTSCR2/PICT1 (Cell Signaling Technology,73225), MDM2 (Santa Cruz, sc-965), RPL11 (Cell Signaling Technology, 18163), RPL5 (Cell Signaling Technology,14568), anti-BrdU (abcam, ab6326), BOP1 (Santa Cruz, sc-390672), RRP1 (GeneTex, GTX115107), UBF (Santa Cruz, sc-13125), FBL (Santa Cruz, sc-374022 and sc-166001), NCL/C23 (Santa Cruz, sc-55486 and Cell Signaling Technology, #14574), NPM1 (Cell Signaling Technology, #3542), Anti-FLAG M2 (Sigma, F3165), p70 S6 Kinase (Cell Signaling Technology, #2708), Phospho-p70 S6 Kinase (Thr389) (Cell Signaling Technology, #97596), and LC3A/B (Cell Signaling Technology, #4108).

## Statistical analysis

Statistical significances were determined using Prism software (GraphPad Software, San Diego, California) as indicated in the figure legends.

## Supporting information

**S1 Fig. Supplemental figures related to Fig 1.** (A) Schema of genome editing for *ACLY*. A GFP-tagged sgRNA/Cas9 plasmid targeting *ACLY* was transfected into HT1080 cells. Following cell sorting by flow cytometry for GFP positive transfected cells, cells were recovered in either standard medium (0-mM added acetate) or 2- or 20-mM sodium acetate–supplemented media. Recovered cell clones were genotyped to identify genome editing for *ACLY* as in "B". (B) Schema of genotyping for *ACLY* genome-edited cell clones and representative genotyping results (top left). The sgRNA-targeted and Cas9 cleavage site in *ACLY* exon 6, which contains a single *Xho*I site overlapped with a Cas9 cleavage site, was PCR amplified by flanked primers (blue). *Xho*I digestion of the PCR product (385 bp) provided 238 bp and 147 bp fragments

when the *Xho*I site was intact (top right). *Xho*I resistance was indicative of insertion or deletion (in/del) by CRISPR-mediated non-homologous end joining (NHEJ). Numbers of clones that exhibited complete (or partial) *Xho*I resistance and numbers of clones screened were shown for each culture media condition (bottom). (C) The sequence of exon 6 in wild-type *ACLY* and the edited sequences in ASA-KO14 and KO17 cells. The sgRNA target site (red), PAM motif (green), Cas9 cut site (arrow head), *Xho*I site, and detected in/del were indicated. We detected only a single pattern of insertion in ASA-KO14 cells. Note that 1 allele of ASA-KO17 possesses an indel, which causes a frame shift mutation replacing amino acids "T-Y-I-E" with "Q". This might explain the detected ACLY peptides in the TMT proteomics using ASA-KO17 cells (S3 Table).
(TIF)

**S2 Fig. Supplemental figures related to Fig 1.** HPLC UV chromatograms of cell lysates from a representative experiment for acetyl-CoA and CoA quantification shown in Fig 1E. Standard acetyl-CoA (100 μM) and CoA (100 μM) run in the same experiment are also shown. Values in parentheses indicate the peak area.
(TIF)

**S3 Fig. Supplemental figures related to Fig 2.** (A) Immunoblotting for the indicated acetylated and total protein levels in ASA-WT and ASA-KO cells cultured in 1% FBS containing media with or without 20-mM acetate for the indicated times. (B) Venn diagram for downregulated transcripts {$\log_2$ FC [(−) Acetate / (+) Acetate] < −1.0. q<0.05} and up-regulated transcripts {$\log_2$ FC [(−) Acetate / (+) Acetate] >1.0. q<0.05} in the RNA sequencing with the 10% and 1% FBS conditions (S2 Table).
(TIF)

**S4 Fig. Supplemental figures related to Fig 3.** (A and B) Immunoblotting for nucleolar protein BOP1 and RRP1 (A) and PICT1 (B) in ASA-WT and ASA-KO cells cultured in 10% or 1% FBS containing media with or without acetate for 4 hours. αTubulin was used as a loading control. (C) Schematic representation of the nucleolus consisting of the fibrillar center (FC; green), the dense fibrillar component (DFC; yellow), and the granular component (GC; gray). (D–F) Immunostaining for FBL (D) and NCL (E and F) along with rRNA staining in ASA-KO (D and E) and ASK-WT (F) cells cultured in 1% FBS containing media with or without acetate for 4 hours. The scale bars under the left images indicate 50 μm. Magnified nuclear images (surrounded by a white square) are shown. Line profiles for indicated fluorescent intensities (FI) determined along the white dashed lines are shown to the right. (G) Quantification of the mean fluorescent intensity of the rRNA signal per nucleus in "F." Data are shown as mean ± SD (*n* = 51 cells per condition). The data underlying the graphs in S4 Fig can be found in S1 Data.
(TIF)

**S5 Fig. Supplemental figures related to Fig 4.** (A) Relative mRNA expressions of p53 target genes selected from the RNA sequencing in S2 Table. Data are shown as mean ± SD (*n* = 3 biological replicates). (B) Immunoblotting for levels of p53, PICT1, γH2A.X, and histone H2A in ASA-KO cells cultured in 10% or 1% FBS containing media with or without acetate for 4 hours, or treated with 5-nM ActD or 1-μM Camptothecin (CPT), a DNA damage inducer, for 4 hours. (C) Immunostaining for γH2A.X in ASA-KO cells cultured in the same conditions as in "A." The scale bars indicate 10 μm. (D) Immunoblotting for levels of phosphorylated Threonine 389 of S6 kinase (S6K), total S6K, LC3, and p53 in ASA-KO cells cultured in 10% or 1% FBS containing media with or without acetate, or treated with 1-μM Torin 1 for 4 hours. The data underlying the graphs in S5 Fig can be found in S1 Data.
(TIF)

**S6 Fig. Supplemental figures related to Fig 5.** (A) Immunoblotting for the indicated proteins in ASA-KO cells cultured in 1% FBS containing media with or without acetate, and in the presence or absence of indicated HDAC inhibitors for 4 hours. (B) Immunoblotting for indicated proteins in ASA-KO cells cultured in 1% FBS containing media with or without acetate, and in the presence or absence of indicated HDAC inhibitors (10 μM) for 4 hours. (C) Immunostaining for NCL along with rRNA dye staining in ASA-KO cells cultured in 1% FBS containing media with or without acetate and in the presence or absence of indicated HDAC inhibitors (50-μM TMP195, 10-μM LMK, or 50-μM Entinostat) for 4 hours. The scale bar under the left images indicates 50 μm. Magnified nuclear images (surrounded by a white square) are shown. Line profiles for indicated fluorescent intensities (FI) determined along the white dashed lines are shown to the right. (D) Quantification of the mean fluorescent intensity of the RNA signal per nucleus in (C). Data are shown as mean ± SD ($n$ = 29 to 53 cells per condition). ****$P$ < 0.0001 (1-way ANOVA followed by Tukey multiple comparisons test). The data underlying the graphs in S6 Fig can be found in S1 Data.
(TIF)

**S7 Fig. Supplemental figures related to Fig 5.** (A) Gene ontology (GO) analysis for TMP195-sensitive 365 acetylated peptides {**S1 Table**, column C, $\text{Log}_2$ [FC: (−) Acetate + TMP195 / (−) Acetate] >0.96}. Enriched representative biological processes are shown. (B) List of selected nucleolar proteins from S1 Table as in "A," and their acetylation sites affected by TMP195. (C) Schematic representation of acetylation sites in NCL (top). TMP195-sensitive acetylation sites are shown in red. RRM, RNA recognition motif. Immunoprecipitation with the acetyl-lysine motif antibody and immunoblotting for NCL-3xFLAG in ASA-KO cells cultured in 1% FBS containing media with or without acetate, and in the presence or absence of 50-μM TMP195 for 90 minutes (bottom). (D) FRAP of NCL-mGFP in ASA-KO cells cultured in 1% FBS containing media with or without acetate, and in the presence or absence of 50-μM TMP195, 10-μM LMK235, or 50-μM Entinostat for 90 minutes. Data are shown as mean ($n$ = 10 independent cells from 2 independent experiments). Representative nucleolar images expressing NCL-mGFP before and after photobleaching are shown on the top. The red allow indicates the ROI. (E) Half recovery times (sec) obtained from the FRAP curves in "D." Data are shown as mean ± SD ($n$ = 10 independent cells). ***$P$ < 0.001, ****$P$ < 0.0001 (1-way ANOVA followed by Tukey multiple comparisons test). The data underlying the graphs in S7 Fig can be found in S1 Data.
(TIF)

**S8 Fig. HCT116 p53-YFP ASA-KO cells.** (A) Immunoblotting for ACLY and p53 levels in HCT116 p53-YFP ASA-WT and ASA-KO cells cultured in 10% or 1% FBS containing media with or without acetate for 4 hours. Histone H3 is shown as a loading control. (B) Immunoblotting for p53 levels in HCT116 p53-YFP ASA-KO cells cultured in 1% FBS containing media with or without acetate, and in the presence or absence of indicated HDAC inhibitors for 4 hours. Following concentrations of HDAC inhibitors were used: 10-μM Vorinostat (VOR), 70-μM Entinostat (ENT), 50-μM TMP195 (TMP), and 10-μM LMK235 (LMK). (C) Immunostaining for NCL along with rRNA dye staining in ASA-KO cells cultured in 1% FBS containing media with or without acetate, and in the presence or absence of indicated HDAC inhibitors (50-μM TMP195, 10-μM LMK, or 50-μM Entinostat) for 4 hours. The scale bar under the left images indicates 50 μm. Magnified nuclear images (surrounded by a white square) are shown. Line profiles for indicated fluorescent intensities (FI) determined along the white dashed lines are shown to the right. The data underlying the graphs in S8 Fig can be found in S1 Data.
(TIF)

**S1 Table. List of peptides identified by acetylome analysis in ASA-KO cells.** $Log_2$ fold changes (FC) in acetylated peptides in ASA-KO cells cultured in 1% FBS containing media with or without acetate (column B), or cultured without acetate in the presence or absence of TMP195 (column C), for 90 minutes are shown (mean of 2 independent technical replicates). (XLSX)

**S2 Table. List of genes identified by RNA sequencing analysis in ASA-KO cells.** $Log_2$ fold changes (FC) in mRNA expression in ASA-KO cells cultured in 10% FBS (sheet 1) or 1% FBS (sheet 2) containing media with or without acetate for 4 hours are shown ($n$ = 3 biological replicates). (XLSX)

**S3 Table. List of proteins identified by Tandem mass tag mass spectrometry analysis in ASA-KO cells.** Fold changes (FC) in protein expression in ASA-KO cells cultured in 1% FBS containing media with or without acetate for 4 hours are shown ($n$ = 3 biological replicates). (XLSX)

**S1 Data. The excel spreadsheet contains the numerical values for Figs 1C, 1D, 1E, 1F, 2A, 2C, 2G, 3A, 3C, 3D, 3E, 3F, S4D, S4E, S4F, S4G, S5A, S5E, S6C, S6D, S7A, S7D, S7E and S8C.** (XLSX)

**S1 Raw images. Original images for blots and gels.** (PDF)

## Acknowledgments

We are grateful to members of Finkel lab for valuable discussion and technical supports and to the NHLBI Cores (Biochemistry Core, Bioinformatics and Computational Biology Core, DNA Sequencing and Genomics Core, Flow Cytometry Core, Light Microscopy Core, and Proteomics Core) and Cores at the University of Pittsburgh (the Health Sciences Metabolomics and Lipidomics Core and the Flow Cytometry Core at the McGowan Institute) for experimental supports. We wish to acknowledge Duck-Yeon Lee at the NHLBI Biochemistry Core for LC-UV instrumentation and calibration for acetyl-CoA and CoA.

## Author Contributions

**Conceptualization:** Toren Finkel, Yusuke Sekine.

**Formal analysis:** Michael J. Calderon, Fayaz Seifuddin, Guanghui Wang, Mehdi Pirooznia, Matthew P. Stokes.

**Funding acquisition:** Shiori Sekine, Jacob Stewart-Ornstein, Stacy G. Wendell, Toren Finkel, Yusuke Sekine.

**Investigation:** Ryan Houston, Shiori Sekine, Michael J. Calderon, Guanghui Wang, Hiroyuki Kawagishi, Daniela A. Malide, Alissa J. Nelson, Matthew P. Stokes, Steven J. Mullett, Stacy G. Wendell, Yusuke Sekine.

**Methodology:** Hiroyuki Kawagishi, Matthew P. Stokes, Jacob Stewart-Ornstein, Yusuke Sekine.

**Project administration:** Yusuke Sekine.

**Resources:** Jacob Stewart-Ornstein.

**Supervision:** Daniela A. Malide, Yuesheng Li, Marjan Gucek, Mehdi Pirooznia, Matthew P. Stokes, Simon C. Watkins, Toren Finkel, Yusuke Sekine.

**Validation:** Yusuke Sekine.

**Visualization:** Shiori Sekine, Yusuke Sekine.

**Writing – original draft:** Yusuke Sekine.

**Writing – review & editing:** Ryan Houston, Shiori Sekine, Michael J. Calderon, Guanghui Wang, Hiroyuki Kawagishi, Yuesheng Li, Mehdi Pirooznia, Alissa J. Nelson, Matthew P. Stokes, Jacob Stewart-Ornstein, Steven J. Mullett, Stacy G. Wendell, Simon C. Watkins, Toren Finkel, Yusuke Sekine.

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
