## [Editor Report · Decision Letter 0]

3 Feb 2020

Dear Dr Sekine, 

Thank you for submitting your manuscript entitled "Acetylation-mediated remodeling of the nucleolus regulates cellular acetyl-CoA responses" for consideration as a Research Article by PLOS Biology.

Your manuscript has now been evaluated by the PLOS Biology editorial staff as well as by an academic editor with relevant expertise and I am writing to let you know that we would like to send your submission out for external peer review.

Please re-submit your manuscript within two working days, i.e. by Feb 05 2020 11:59PM.

Kind regards,

Lauren A Richardson, Ph.D

Senior Editor

PLOS Biology

---

## [Decision Letter · Decision Letter 1]

9 Apr 2020

Dear Dr Sekine,

Thank you very much for submitting your manuscript "Acetylation-mediated remodeling of the nucleolus regulates cellular acetyl-CoA responses" for consideration as a Research Article at PLOS Biology. Thank you also for your patience as we completed our editorial process, and please accept my sincere apologies for the long delay in providing you with our decision. Your manuscript has been evaluated by the PLOS Biology editors, an Academic Editor with relevant expertise, and by two independent reviewers. We were expecting comments from another two reviewers, but they have not responded to any of our chases.

As you will see, the reviewers are positive of the study and find the conclusions interesting and worth considering for publication. Nevertheless, the reviewers also find that some of the results are not sufficiently supported by the experiments and suggest additional work to strengthen the results. After discussing the reviews with the Academic Editor, we do feel that points by Reviewer 2 about validating the acCoA quantitative data and assessing mTOR signalling are important and should be addressed. Also, as mentioned by both reviewers, the statement 'Overall, these observations demonstrate that class IIa HDACS-mediated deacetylation alters the dynamics of nucleolar proteins,' requires further support or toning down. While addressing the other points would strengthen the conclusions, we do not feel they are essential for publication.

In light of the reviews (attached below), we will not be able to accept the current version of the manuscript, but we would welcome re-submission of a revised version that takes into account the reviewers' comments. We cannot make any decision about publication until we have seen the revised manuscript and your response to the reviewers' comments. Your revised manuscript is also likely to be sent for further evaluation by the reviewers.

We expect to receive your revised manuscript within 3 months. Please email us (plosbiology@plos.org) if you have any questions or concerns, or would like to request an extension - we do understand that you might not have access to your lab for a while due to COVID-19. At this stage, your manuscript remains formally under active consideration at our journal; please notify us by email if you do not intend to submit a revision so that we may end consideration of the manuscript at PLOS Biology.

**IMPORTANT - SUBMITTING YOUR REVISION**

*Re-submission Checklist*

*Published Peer Review*

*PLOS Data Policy*

*Blot and Gel Data Policy*

Sincerely,

Ines

--

Ines Alvarez-Garcia, PhD

Senior Editor

PLOS Biology

Carlyle House, Carlyle Road

Cambridge, CB4 3DN

+44 1223–446970

Reviewers’ comments

Rev. 1:

The manuscript by Dr. Sekine and colleagues, entitled "Acetylation-mediated remodeling of the nucleolus regulates cellular acetyl-CoA responses" clearly demonstrate that decrease in acetyl-CoA levels causes deacetylation of a significant number of nucleolar proteins in a class IIa HDACs deacetylate dependent manner, which compromises rRNA synthesis and activated p53. They have also found that acetyl-CoA decreasing remodels the nucleolar structure via regulation the properties of the LLPS of the nucleolus.

Their data is clean and straightforward, the text is clear, and the conclusions are well supported by the experiments. There are, however, some experimental aspects that should addressed before I would consider this manuscript ready for publication in PLOS BIOLOGY.

Major points:

Fig. 2D and E: I recommend that authors perform the same combination of experiments using ASK-WT cells.

Fig. 5: Authors have shown that the class IIa HDAC family proteins regulate nucleolar protein deacetylation and nucleolar dynamics by the experiments using specific inhibitors. It would be great if they could specify which of class IIa HDAC family regulates these process by KD experiment using siRNAs.

Fig. 5G. From this FLAP analysis, authors conclude that class IIa HDACs-mediated deacetylation regulates the dynamics of nucleolar proteins, which are likely associated with change in the LLPS status. The observation is very important findings. However, they have tested only one nucleolar protein, NPM1.

I consider that that they should test at least another nucleolar protein (e.g. FBL, a DFC marker protein, or NCL, a GC marker protein).

Minor points:

Pages 16-17:

They discuss that "Once acetyl-CoA levels fall, class IIa HDACs deacetylate a significant number of nucleolar proteins. These reactions remodel the properties of the LLPS of the nucleolus, resulting in impairment of rRNA synthesis and induction of the p53-mediated stress responses".

However, we could not rule out the possibility that "Once acetyl-CoA levels fall, class IIa HDACs deacetylate a significant number of nucleolar proteins. These reactions result in impairment of rRNA synthesis, which in turn remodels the properties of the LLPS of the nucleolus and changes the nucleolar structure."

I think they should also consider this possibility.

Page 12, line 23:

"HDM2" instead of "MDM2" because they used human cells (HT1080).

Page 13, line 1:

"protein stability" instead of "protein expression"

Page 17, line 1:

"nucleolus" instead of "nucelolus"

Rev. 2:

The manuscript by Houston et al. explores the relationship between acetyl-CoA biosynthesis, protein acetylation, and nucleolar function. Specifically, they develop a human fibrosarcoma cell line model in which ATP-citrate lyase (ACLY) is knocked out, and require acetate for proliferation. Acetyl-CoA levels are lower in the knockout cell line upon acetate withdrawal, which the authors use to explore the downstream consequences of depletion of this metabolite. Interestingly, they find that acute depletion of acetyl-CoA does not appear to greatly alter lipid or cholesterol levels; instead, gene expression, proteomic, and acetyl-proteome profiling lead them to identify a depletion in the levels and acetylation of nucleolar proteins. A series of mechanistic follow-up studies are performed, which provides evidence that withdrawal of acetate from ACLY-knockout cells reduces rRNA biogenesis and induces RPL11/RPL5-dependent p53 activation in a manner that can be rescued by HDAC inhibitors, implying a role for protein acetylation in this process. Finally, it is shown acetate also preferntially affects nucleolar liquid-liquid phase separation in ACLY-knockout cells, as assessed by FRAP studies of NCL-mGFP. Overall, the experiments are interesting and well done. If sufficient evidence is provided to implicate acetyl-CoA levels in these phenomena, it will be of substantial interest to the PLOS Biol readership and biological community. However, there are a few experiments missing that must be performed in order to better support some of the major claims of the paper:

1. Quantification of acetyl-CoA: The basis for all of the biological results in the manuscript is that acetate withdrawal from the ACLY knockout cells alters acetyl-CoA levels. This is demonstrated in Figure 1E of the main text, which is quantified on the basis of an HPLC assay examining acetyl-CoA levels from ~2M cells according to the Supplemental. Acetyl-CoA (and CoAs in general) are remarkably difficult metabolites to quantify, suffering both from variable extraction efficiency due to their hydrophobicity as well as lability of their 3'-phosphate group. Based on the figure, the author's measurements of acetyl-CoA and CoA in the acetate-depleted a ratio of ~1:50, which is far beyond what has been measured in previous studies by LC-MS. Since every claim in the paper hinges on the validation of this system to deplete acetyl-CoA, it is absolutely essential that the authors provide an orthogonal measure (i.e. using LC-MS) to demonstrate the dynamics of acetyl-CoA in their system.

2. Given the connection between acetyl-CoA and the nucleolus, it is also essential that the authors also describe what mTOR signaling is doing in these cells. mTor is the most well-known transducer of signals regarding cellular metabolism to ribosome biogenesis. It is certainly conceivable that depletion of acetyl-CoA could cause mTOR inactivation, and previous studies (which should be cited) have linked acetyl-CoA to mTOR's activation state: https://www.ncbi.nlm.nih.gov/pubmed/30197302. The authors should perform Western blotting examining the status of the mTOR pathway using conventional measures (S6 kinase phosphorylation, etc).

3. Page 15: Where on the RPL11 protein does acetylation take place (i.e. what lysine residue)? Based on previous studies, what is the mechanistic hypothesis for how acetylation functionally promotes its activity and impedes activation of p53. Please cite any ltierature precedent.

4. I found the liquid-liquid phase separation section of the manuscript to be speculative and uncompelling. The changes in FRAP signal appear to be minor and certainly lack context (i.e. comparison to a stimuli predicted to have a 'major' effect) and the connection to the rest of the manuscript was unclear. This seems like an interesting preliminary experiment for a future paper rather than one essential to the current results, and I would suggest moving it to the SI or removing it to increase focus/clarity,.

5. With the above comment (#4), the statement, 'Overall, these observations demonstrate that class IIa HDACS-mediated deacetylation alters the dynamics of nucleolar proteins,' is far too strong a statement based on the data. This should be re-phrased to indicate the current state of the data, 'Overall these observations are consistent with a model in which class IIa HDACS-mediated deacetylation may alter the dynamics of nucleolar proteins, which has the potential to alter the phase state of the nucleolus."

Minor points:

1. The strategy described for acetyl-CoA modulation based on ACLY knockout and acetate supplementation is entirely derivative of one that was published by Wellen et al. (Cell Rep, 2016). This was performed for both MEFs as well as glioblastoma cells. The authors mention it, but I found the citation inappropriate as they imply this has not been done previously in human cells by mentioning only the MEF work (page 7). It would not hurt the novelty of this study to more properly acknowledge the field.

2. Consistent with the discussion of acetyl-CoA/CoA quantification above, the authors should provide HPLC chromatograms and calibration curves for the metabolite quantifications given in Figure 1E. This is important as again, the entire manuscript hinges on the quality of these quantificaitons.

3. The detection of dynamically acetylated ribosomal proteins is interesting; however, proteomic datasets are biased towards the detection of ribosomal proteins as they are the most abundant proteins in cells, and they may be modified at low stoichometries that are unlikely to affect their activity (https://www.nature.com/articles/s41467-019-09024-0). The authors should comment on this issue, and also explicitly describe if any of their acetyl-CoA/dynamic acetylations are found at 'high stoichiometry sites as described by Hansen et al. (referenced in link above)

---

## [Decision Letter · Decision Letter 2]

13 Oct 2020

Dear Dr Sekine,

Thank you for submitting your revised Research Article entitled "Acetylation-mediated remodeling of the nucleolus regulates cellular acetyl-CoA responses" for publication in PLOS Biology. Thanks again for your patience as we completed our editorial process. I have now obtained advice from one of the original reviewers and have discussed his/her comments with the Academic Editor. The academic editor has also checked your responses to the other reviewers.

We're delighted to let you know that we're now editorially satisfied with your manuscript. However before we can formally accept your paper and consider it "in press", we also need to ensure that your article conforms to our guidelines. A member of our team will be in touch shortly with a set of requests. As we can't proceed until these requirements are met, your swift response will help prevent delays to publication. Please also make sure to address the data and other policy-related requests noted at the end of this email.

- a cover letter that should detail your responses to any editorial requests, if applicable

*Copyediting*

*Published Peer Review History*

*Early Version*

Sincerely,

Ines

--

Ines Alvarez-Garcia, PhD,

Senior Editor,

ialvarez-garcia@plos.org,

PLOS Biology

DATA POLICY:

Thank you for providing all the data underlying the graphs shown in the figures. Please also ensure that figure legends (both main and supplementary) in your manuscript include information on WHERE THE UNDERLYING DATA CAN BE FOUND.

Reviewer's comments

Rev. 1:

The authors improved manuscript in response to my comment. Though they have not done all the experiments that I requested, their response to my comments is reasonable. Therefore, I consider that this manuscript should be accepted.

Only one minor point:

LMK235 instead of LMK234 (page 18, line 8).

---

## [Editor Report · Decision Letter 3]

29 Oct 2020

Dear Dr Sekine,

On behalf of my colleagues and the Academic Editor, Katy Wellen, I am pleased to inform you that we will be delighted to publish your Research Article in PLOS Biology. 

PRODUCTION PROCESS

Before publication you will see the copyedited word document (within 5 business days) and a PDF proof shortly after that. The copyeditor will be in touch shortly before sending you the copyedited Word document. We will make some revisions at copyediting stage to conform to our general style, and for clarification. When you receive this version you should check and revise it very carefully, including figures, tables, references, and supporting information, because corrections at the next stage (proofs) will be strictly limited to (1) errors in author names or affiliations, (2) errors of scientific fact that would cause misunderstandings to readers, and (3) printer's (introduced) errors. Please return the copyedited file within 2 business days in order to ensure timely delivery of the PDF proof. 

If you are likely to be away when either this document or the proof is sent, please ensure we have contact information of a second person, as we will need you to respond quickly at each point. Given the disruptions resulting from the ongoing COVID-19 pandemic, there may be delays in the production process. We apologise in advance for any inconvenience caused and will do our best to minimize impact as far as possible.

EARLY VERSION

PRESS 

Kind regards,

Alice Musson

Publishing Editor, 

PLOS Biology

on behalf of

Ines Alvarez-Garcia,

Senior Editor

PLOS Biology